# Is it Bigger than a Breadbox: Efficient Cardinality Estimation for Real World Workloads

## Abstract

DB engines produce efficient query plans by relying on cost models, that estimate cardinalities of subqueries constituting the input query. Standard cardinality estimators rely on heuristics, with magic numbers tuned to improve average performance on benchmarks. Empirically, their estimation error increases with query complexity. Recently, learning-based neural estimators have been proposed. However, they are not adopted in-practice, due to added operational complexity. Training or tuning generally requires extracting or simulating training data, from database contents or sample queries. Rather than learning one monolithic network, we are motivated to capture the common practical scenario: query patterns repeatedly re-appear soon after their first introduction. We propose LITECARD, that online-learns many (simple) regressors, one per subquery pattern. The regressor corresponding to a pattern can be randomly-accessed using hash of directed acyclic graph encoding the subquery. LITECARD instantiates from cold-start, imposing negligible overhead, while competing with SoTA neural approaches on error metrics. More importantly, implementing LITECARD into PostgreSQL reduces query execution time by 30% of popular JOB-lite workload on IMDb, while incurring negligible overhead (37 seconds) for online learning. By achieving notable accuracy and runtime improvements over standard methods, and drastically reduces operational costs compared to neural estimators, LITECARD pushes Pareto frontiers balancing *execution speedups* and *overhead time*.

## 1 Introduction

The majority of computer applications of any significant utility use relational databases. Performance optimization of query execution has therefore been studied for decades, *e.g.*, Astrahan et al. (1976); Selinger et al. (1979); Graefe & DeWitt (1987); Ioannidis et al. (1997); Trummer & Koch (2015). **Cardinality Estimation** – the task of predicting the record-count of (sub-)queries – is essential for query plan optimization (Leis et al., 2015; Marcus et al., 2021; Lee et al., 2023).

The popular database engine, PostgreSQL, estimates cardinalities using per-column histograms (PostgreSQL Group, 2025), naïvely assuming that columns are uncorrelated. Advantages of this heuristic include its speed-of-calculation, which allows it to be invoked numerous times for multi-join queries. However, this estimation exhibits large errors when independence assumptions are violated, *e.g.*, when joining records from multiple tables, unnecessarily slowing-down query execution by possibly orders-of-magnitudes (Moerkotte et al., 2010).

A variety of deep-learning methods propose to capture intricate data distributions, either directly by sampling records (*e.g.*, Hilprecht et al., 2020; Wu et al., 2023), or indirectly by posing *cardinality estimation* as a supervised learning task (*e.g.*, Kipf et al., 2019; Chronis et al., 2024). While these models can discover correlations across columns and produce better cardinality estimates than heuristic algorithms, their overheads prevents their adoption in practice (Wang et al., 2021).

In this paper, we strive to design a cardinality estimator that: (i) can run from cold-start, requiring no upfront training; (ii) can adapt to changes in workloads or data shifts; and (iii) has negligible update and inference time. We propose such an estimator. Rather than a monolithic neural network that processes all queries, we employ many small models, each specializes to one sub-query pattern. The query pattern is identified from the *structure* of the graph corresponding to the query, while

excluding some node features, *e.g.*, constant values, table names and/or column names. Our proposed method fits within a general a class of learning methods known as *locally-weighted models*. Prediction on any data point requires fitting a new model on training examples that are near the data point. These methods define a (similarity) *Kernel* function, that generally operates on pairs of **numeric** feature vectors. However, our kernels integrate **both** the **graph structure and numeric** data. Crucially, specializing online to emerging query patterns: van Renen et al. (2024) show that >95% of queries repeat in same template within a month.

Our **main contributions** are as follows. We are the first to propose an online-learning cardinality estimator that is invariant to many SQL transformations. The crux of our method relies canonical ordering of nodes in a DAG: the feature vector is invariant to node renumbering (§3.2). This canonicalization allows learning many simple regressors, that we store in a hierarchy data structure, divisively partitioning all executed subqueries online (§3.5). The data structure can be used (§3.4) to provide accurate cardinality estimates, without needing to be trained apriori, allowing Postgres to arrive at query plans that execute 30% fast in practice.

## 2 BACKGROUND

### 2.1 GRAPH REPRESENTATION OF (SUB)QUERIES AND QUERY PLAN OPTIMIZATION

Database engines rely on *cost models* to create efficient *query execution plan* for responding to a query. The plan is a tree: leaf-nodes read data records, generally from table columns, and as the data traverses down the tree, records get merged (per joined columns) and filtered (per predicates), finally producing one record stream at the root, *i.e.*, the response to the query. There can be many valid plans for a query. However, some plans are favored, requiring fewer resources and executing faster. While searching for an optimal plan, the cost model must estimate the cardinality of candidate sub-queries (nodes) before they get selected into the query plan (tree). The cardinality is the number of records output by the subquery (emitted by the node, down the tree). Consider the simple SQL:

**SELECT ... FROM** movies **WHERE** stars>3 and year IN (2024,2025)  (1)

The statement queries movies produced in the last 2 years, rated above 3-stars. Let us assume that both columns, `stars` and `year`, are individually indexed but are not co-indexed. Then, the Query Plan Optimizer estimates the cardinality of two constituent sub-queries:

**SELECT...WHERE** stars > 3   and   **SELECT...WHERE** year IN (2024, 2025)

The optimizer uses cardinality estimates to determine *join types*. For instance, if the second subquery has a low cardinality estimate, then it could be executed earlier, and its (primary-key, record) outputs can be stored in-memory before the first subquery executes. However, if both subqueries have large cardinalities, then they can be separately executed, sorted by primary key, then intersected in a streaming-fashion. These are respectively named *broadcast join* and *merge join*. Cardinalities also determine *join orders*. For instance, when joining 3 tables (A⋈B⋈C), the optimizer must choose which two tables merge first ((A⋈B)⋈C) or (A⋈(B⋈C)). The number of join orderings can be exponential in the number of tables. While searching for the optimal plan, the optimizer repeatedly invokes the cardinality estimator, *e.g.*, up to thousands of times for complex queries.

**Graph Representation of (sub)queries.** Queries are generally represented as trees in database engines (Pirahesh et al., 1992; Liu & Özsu, 2018; Ramakrishnan & Gehrke, 2003), and we convert them to directed acyclic graphs (DAGs) similar to Chronis et al. (2024). Details are in appendices A&D. Figure 1 depicts a DAG corresponding to Eq. 1. There are different node types (colored), and each type has its own feature sets. Let $\mathcal{T}$ denote the universe of node types that can appear in the (sub)query graph. In our application,

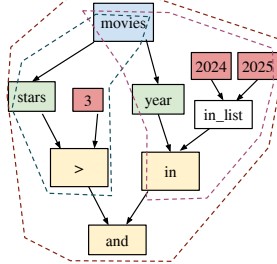

$$\mathcal{T} = \{\underbrace{table, alias, column, literal, op, function}_{\text{for graphs extracted from SQL or PostgreSQL's RelInfo}}, \underbrace{join, scan, ..}_{\text{for PostgreSQL's}}\} \quad (2)$$

Figure 1: SQL DAG. Planner estimates cardinalities of dash-marked subqueries.

For algorithmic correctness, all sets {.} are ordered. Let $\mathcal{A}$ be set of pairs (type, attribute name):

$$\mathcal{A} = \{(table, name), (column, name), (column, type), (literal, value), (op, code), \dots\} \quad (3)$$

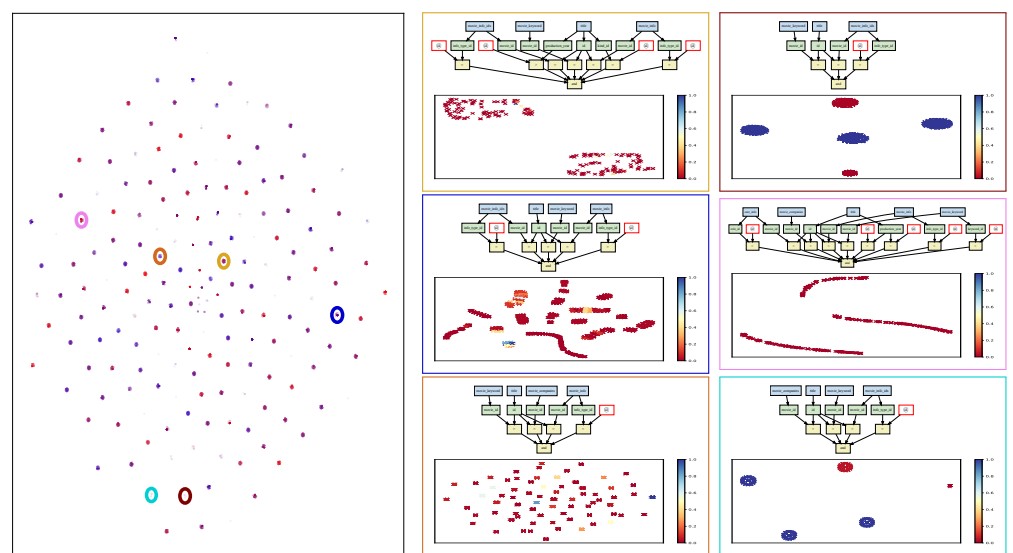

Figure 2: t-SNE visualizations of IMDB 5K workload. **(Left)** Every subquery is a point (with 5% opacity). Due to $K_{\mathcal{F}}^{\mathcal{H}}(G, G') = \mathbf{1}_{[h^{\mathcal{H}}(G)=h^{\mathcal{H}}(G')]} \times .$, subquery DAGs that are isomorphic (per $\mathcal{H}$) are cleanly clustered, painting a darker region. The point color represents cardinality of the query (from red to blue). We choose 6 clusters (by stratified sampling) and circle them with colors. **(Right)** we recompute t-SNE **within** each colored cluster. The original dimension of every right plot equals the number of $\boxed{@}$ nodes in the graph above it, which renders the subquery pattern graph. Finally, points are colored using their ground-truth (normalized) cardinalities.

## 2.2 LOCALIZED MODELS

**Local models** infer on a data point $\mathbf{x}$ by considering **nearby** points. Proximity between points $\mathbf{x}$ and $\mathbf{z}$ is measured by kernel function $K(\mathbf{x}, \mathbf{z}) \geqslant 0$. A notable choice is the Gaussian kernel with

$$K_\sigma(\mathbf{x}, \mathbf{z}) = \exp\left(-\frac{||\mathbf{x} - \mathbf{z}||^2}{\sigma^2}\right) \in [0, 1] \qquad (4)$$

where hyperparameter $\sigma > 0$ is known as the *kernel width* or *variance*. This kernel frequently appears. We utilize it in two ways. First, in *locally-weighted linear regression* (Cleveland, 1979), Second, in one-shot prediction (Hechenbichler & Schliep, 2004).

## 3 GRAPH-LOCAL LEARNING

We first present our final model, top-to-bottom, and the remainder of the section provides details.

Let $(\mathcal{G}', y') \in \mathcal{D}$ denote history of previously-seen (sub)query DAGs, each associated with its cardinality. History $\mathcal{D}$ starts empty and populates while queries are executing. Fig. 1 captures three such DAGs, each rooted at a yellow node.

Inspired by §2.2, given a test (sub)query graph $\mathcal{G}$, we estimate its cardinality by inference:

$$g_\theta(\mathcal{G}) \quad \text{where} \quad \theta = \arg\min_{\theta'} \sum_{(\mathcal{G}', y') \in \mathcal{D}} K_{\mathcal{F}}^{\mathcal{H}}(\mathcal{G}, \mathcal{G}') \times (g_{\theta'}(\mathcal{G}') - y')^2, \qquad (5)$$

The hyperparameters **pattern features** $\mathcal{H} \subset \mathcal{A}$ and **learning features** $\mathcal{F} \subset \mathcal{A}$ are explained in §3.2. Kernel $K_{\mathcal{F}}^{\mathcal{H}}(., .) \geqslant 0$ outputs large value if its inputs are similar, both feature- and structure-wise, as:

$$K_{\mathcal{F}}^{\mathcal{H}}(\mathcal{G}, \mathcal{G}') = \underbrace{\mathbf{1}_{[h^{\mathcal{H}}(\mathcal{G})=h^{\mathcal{H}}(\mathcal{G}')]}}_{G \& G' \text{ are isomorphic}} \times \underbrace{K_\sigma\left(\mathbf{x}_{\mathcal{F}}^{\mathcal{H}}(\mathcal{G}), \mathbf{x}_{\mathcal{F}}^{\mathcal{H}}(\mathcal{G}')\right)}_{\text{their features are nearby}} \qquad (6)$$

where $K_\sigma$ is defined in Eq. 4 and $\mathbf{x}_{\mathcal{F}}^{\mathcal{H}}(\mathcal{G})$ denotes a feature vector containing features listed in $\mathcal{F}$ from $\mathcal{G}$'s nodes, respecting canonical node-ordering established by $\mathcal{H}$. Indicator function $\mathbf{1}_{[h^{\mathcal{H}}(\mathcal{G})=h^{\mathcal{H}}(\mathcal{G}')]}$ evaluates to 1 when $\mathcal{G}$ and $\mathcal{G}'$ are isomorphic when considering features $\mathcal{H}$, and to 0 otherwise.

The model $g_\theta$ is fit locally around $\mathcal{G}$. We restrict ourselves to simple models that can quickly train with negligible overheads. We experiment with Locally-weighted Linear Regression $g_\theta^{\text{LR}}$, in addition to Gradient-boosted Decision Forests $g^{\text{DF}}$ (we use implementation of (Guillame-Bert et al., 2023)). For conciseness, we ignore the regularization terms from Eq. 5, such as $\ell_2$ regularization for Linear Regression, or height-restriction for Decision Forests. Furthermore, we experiment with one-shot predictors following Hechenbichler & Schliep (2004), with:

$$g^{\text{RBF}}(\mathcal{G}) = \frac{1}{Z} \sum_{(\mathcal{G}',y')\in\mathcal{D}} K_{\mathcal{F}}^{\mathcal{H}}(\mathcal{G},\mathcal{G}') \times y' \ \text{ with } \ Z = \sum_{(\mathcal{G}',y')\in\mathcal{D}} K_{\mathcal{F}}^{\mathcal{H}}(\mathcal{G},\mathcal{G}') \tag{7}$$

**System Integration.** We implement functions $g(.)$ and $K_{\mathcal{F}}^{\mathcal{H}}(.,.)$ in open-source PostgreSQL (details are in §D). The Query Planner invokes them while searching for the optimal plan. Once the plan is finalized then executed, cardinalities of all subgraphs (yellow nodes of Fig. 1) are recorded in $\mathcal{D}$.

## 3.1 DEFINITIONS

Let $\{0,1\}^k$ be a string with $k$ bits and let $\{0,1\}^*$ be a string with arbitrary length. We denote a (cryptographic) 256-bit hash \$ : $\{0,1\}^* \to \{0,1\}^{256}$. Let $\mathcal{G} = (\mathcal{V},\mathcal{E},\mathcal{X})$ represent a query graph (depicted in Fig. 1), with node set $\mathcal{V} = \{1,2,\ldots,n\}$ where $n$ denotes number of nodes ($n=10$ in Fig. 1). Edge set $\mathcal{E} \subset \mathcal{V} \times \mathcal{V}$ contains directed edges ($|\mathcal{E}|$=10 in Fig. 1) that must necessarily induce a *directed acyclic graph* (DAG). Reverse edge set $\mathcal{E}^\top = \{(v,u)\}_{(u,v)\in\mathcal{E}}$. The feature set $\mathcal{X} \in (\mathcal{A} \mapsto \{0,1\}^*)^n$ stores multiple features per node. $\mathcal{X}_j[(t,a)]$ denotes accessing string-valued attribute $(t,a) \in \mathcal{A}$ for node $j \in \mathcal{V}$. Suppose $(t,a) = (table,name)$ and $j$ corresponds to the index of blue node of Fig. 1, then $\mathcal{X}_j[(t,a)] = $ "movies". If node $j$ does not have attribute $(t,j)$ then $\mathcal{X}_j[(t,a)]$ defaults to null (or empty-string). Let $\tau_j \in \mathcal{T}$ denote the type of node $j \in \mathcal{V}$.

## 3.2 CANONICAL NODE ORDERING, HASHING AND FEATURE EXTRACTION

**Canonical Node Ordering and Pattern Hashing.** $\mathcal{H} \subset \mathcal{A}$ can effectively partition incoming queries online. We first assemble an array of strings $\mathbf{H} \in \{0,1\}^{n\times 256}$ with row $j \in \mathcal{V}$ initialized as:

$$\mathbf{H}_j^{\mathcal{H}} := \$(\oplus\{\mathcal{X}_j[(t,a)] \mid \tau_j = t\}_{(t,a)\in\mathcal{H}}) \tag{8}$$

where $\oplus\{.\}$ denotes string-concatenation of elements in ordered set $\{.\}$. The hash value $\mathbf{H}_j^{\mathcal{H}} \in \{0,1\}^{256}$ at this initialization $\approx$uniquely[1] identifies node $j$'s feature values, while restricting to pattern features $\mathcal{H}$. Then, we update the entries:

$$\mathbf{H}_j^{\mathcal{H}} := \$\left(\mathbf{H}_j^{\mathcal{H}} \oplus \texttt{sort}(\{\mathbf{H}_k^{\mathcal{H}} \mid (k,j)\in\mathcal{E}\})\right) \ \forall j \in \texttt{TopologicalOrder}(\mathcal{E}), \text{ then }, \tag{9}$$

$$\mathbf{H}_j^{\mathcal{H}} := \$\left(\mathbf{H}_j^{\mathcal{H}} \oplus \texttt{sort}(\{\mathbf{H}_k^{\mathcal{H}} \mid (k,j)\in\mathcal{E}^\top\})\right) \ \forall j \in \texttt{TopologicalOrder}(\mathcal{E}^\top). \tag{10}$$

The array $\mathbf{H}^{\mathcal{H}}$ provides two benefits. First, it uniquely identifies the (sub)query pattern when including only the features in $\mathcal{H}$, used below to define graph-level string $h^{\mathcal{H}} \in \{0,1\}^{256}$. Second, it establishes a canonical ordering $\pi^{\mathcal{H}}$ on $\mathcal{V}$. The hash of a (sub)query pattern (given $\mathcal{H}$) is defined as:

$$h^{\mathcal{H}} = \$\left(\bigoplus_{j\in\pi^{\mathcal{H}}} \mathbf{H}_j^{\mathcal{H}}\right), \quad \text{ with } \quad \pi^{\mathcal{H}} = \arg \texttt{sort}(\{\mathbf{H}_j^{\mathcal{H}}\}_{j\in\mathcal{V}}). \tag{11}$$

**Feature Extraction.** Our framework allows configuring feature extractors, each extractor function $f : \{0,1\}^* \to \mathbb{R}^{d_f}$ converts string features for one node, into a numerical vector of $d_f$ dimensions. We program simple feature extractors that we list in Appendix F. We now introduce our most-important object. Let feature vector $\mathbf{x}_{\mathcal{F}}^{\mathcal{H}}$ contain features of nodes extracted from graph using $\mathcal{F}$, while using the canonical node ordering induced by $\pi^{\mathcal{H}}$. Formally:

$$\mathbf{x}_{\mathcal{F}}^{\mathcal{H}} = \bigoplus_{j\in\pi^{\mathcal{H}}} \left\{f_{(t,a)}(\mathcal{X}_j[(t,a)]) \ \mid \ t = \tau_j\right\}_{(t,a)\in\mathcal{F}}. \tag{12}$$

---

[1]If we assume \$ is a uniform cryptographic hash function, then expected collision rate $\approx \frac{\text{UniqPatterns}}{2^{256}}$.

Table 1: Features used for hashing and model invocation. The choices $\mathcal{H}_1 \subset \mathcal{H}_2 \subset \mathcal{H}_3$ to divisively partition subqueries, forming a hierarchy, as depicted in Fig. 10.

| $k$ | $\mathcal{H}_k$ | $\mathcal{F}_k$ |
|---|---|---|
| 1 | $\mathcal{H}_1 = \{(table, name), (column, type)\}$ | $\mathcal{F}_1 = \mathcal{F}_2 \cup \{(column, numUniques)\}$ |
| 2 | $\mathcal{H}_2 = \mathcal{H}_1 \cup \{(column, name)\}$ | $\mathcal{F}_2 = \mathcal{F}_3 \cup \{(op, code)\}$ |
| 3 | $\mathcal{H}_3 = \mathcal{H}_2 \cup \{(op, code)\}$ | $\mathcal{F}_3 = \{(literal, value)\}$ |

For completeness, the dimensionality of $\mathbf{x}_{\mathcal{F}}^{\mathcal{H}}$ is given by $\sum_{(t,a)\in\mathcal{F}} \sum_{j\in\mathcal{V}} \mathbf{1}_{[t=\tau_j]} d_{f_{(t,a)}}$. It is important to note that the dimensionality of $\mathbf{x}_{\mathcal{F}}^{\mathcal{H}}$'s from two different (sub)query graphs, will be equal if the two graphs have the same number of nodes for every node type $t \in \mathcal{T}$. Theorems 2&3 have details.

Objects $\mathcal{F}$ and $\mathcal{H}$ are configurations and not functions of any particular query graph $\mathcal{G}$. In contrast, the objects $\mathbf{x}_{\mathcal{F}}^{\mathcal{H}}$, $\pi^{\mathcal{H}}$, $\mathbf{H}^{\mathcal{H}}$, and $h^{\mathcal{H}}$ are functions of the input $\mathcal{G}$ and should've written as $\mathbf{x}_{\mathcal{F}}^{\mathcal{H}}(\mathcal{G})$, *etc.*

### 3.3 Correctness Analysis

We establish three theorems and present their ideas. The first two guarantee consistency within any graph, while the last enables learning across graphs. Formal theorems and proofs are in Appendix E.

**Theorem Idea 1** *Any feature set $\mathcal{H} \subseteq \mathcal{A}$ can induce a canonical node ordering.*

**Theorem Idea 2** *The sets $\mathcal{H} \subseteq \mathcal{A}$ and $\mathcal{H} \subseteq \mathcal{A}$ can extract a canonical feature vector.*

**Theorem Idea 3** *Given an arbitrary anchor graph $\mathcal{G}$, then every $\mathbf{x} \in \{\mathbf{x}_{\mathcal{F}}^{\mathcal{H}}(\mathcal{G}') \mid h(\mathcal{G}) = h(\mathcal{G}')\}$ has the same dimensionality, with canonical node-to-feature positions.*

### 3.4 Efficient Online Algorithm

Inference on test $\mathcal{G}$ *seems* inefficient due to summation over history $\mathcal{D}$ (Eq. 5 & 7), however, our choice of $K_{\mathcal{F}}^{\mathcal{H}}$ (Eq. 6) allows random-access lookup of $\{(\mathcal{G}', y) \mid h^{\mathcal{H}}(G) = h^{\mathcal{H}}(G')\}_{(\mathcal{G}',y)\in\mathcal{D}} \triangleq \mathcal{D}_{\mathcal{G}}^{\mathcal{H}}$. In particular, we store in-memory `HashTable` $: h^{\mathcal{H}}(G) \mapsto \{(\mathbf{x}_{\mathcal{F}}^{\mathcal{H}}(\mathcal{G}'), y')\}_{(\mathcal{G}',y')\in\mathcal{D}}$. In fact, we never keep $\mathcal{D}$ in memory. After subquery $\mathcal{G}$ is executed, we append its feature vector $\mathbf{x}_{\mathcal{F}}^{\mathcal{H}}(\mathcal{G})$ and its cardinality onto `HashTable`$[h^{\mathcal{H}}(G)]$ then discard $\mathcal{G}$ to reduce memory footprint. It is possible to further improve the efficiency in multiple ways. For instance, avoid frequent model fitting for $g^{\text{DF}}$ and $g^{\text{LR}}$ (Eq.5), *e.g.*, by storing model parameters, or use approximate nearest neighbors for $g^{\text{RBF}}$ (Eq.7). However, further optimizations are outside the context of this paper, as our setup suffices for our experiments, already speeding IMDb 5k workload by 30% with negligible total overhead time of <40 seconds.

### 3.5 Hierarchical Data Structure

Rather than one choice for each of $(\mathcal{H}, \mathcal{F})$, we include three $\{(\mathcal{H}_1, \mathcal{F}_1), (\mathcal{H}_2, \mathcal{F}_2), (\mathcal{H}_3, \mathcal{F}_3)\}$ and particularly choose $\mathcal{H}_1 \subset \mathcal{H}_2 \subset \mathcal{H}_3$, as listed in Table 1. The choice of $\mathcal{H}'s$ recursively partitions subqueries into a hierarchy of three levels, yielding a data-structure depicted in 10. $\mathcal{H}_1$ is the most general. As visualized in Fig. 10, $h^{\mathcal{H}_1}$ hashes subquery graphs to the same hash value, even though they differ on the op-code or the column name. Then, $h^{\mathcal{H}_2}$ partitions those by column. Finally, $h^{\mathcal{H}_3}$ partitions those by op-code. For inference, we trust the most-specialized model with sufficient observations. Specifically, if $|\mathcal{D}_{\mathcal{G}}^{\mathcal{H}_3}| \geq \beta_3$, then inference is done using the model associated with `HashTable`$\left[h^{\mathcal{H}_3}(G)\right]$, else if $|\mathcal{D}_{\mathcal{G}}^{\mathcal{H}_2}| \geq \beta_2$, then using `HashTable`$\left[h^{\mathcal{H}_2}(G)\right]$, else if $|\mathcal{D}_{\mathcal{G}}^{\mathcal{H}_1}| \geq \beta_1$, then using `HashTable`$\left[h^{\mathcal{H}_1}(G)\right]$, else, then using the traditional cost estimator.

## 4 Experimental Evaluation

We conduct a number of experiments to quantitatively compare our method, LiteCard, against a variety of established baselines, on workloads summarized in Table 2. Our experiments focused on:

- **Run-time performance**. Query planner searches for an optimal query plan while assuming that the cardinality estimates are ground-truth. In general, more accurate cardinality

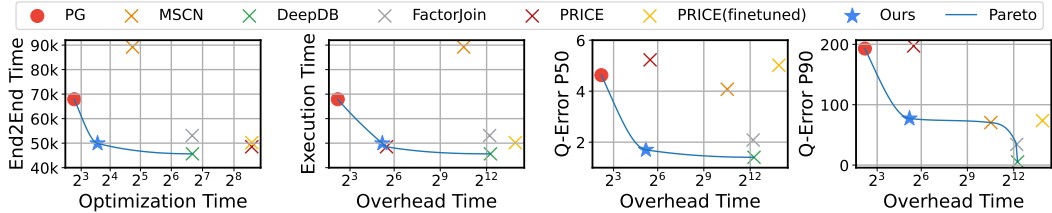

Figure 3: Comparing different techniques on the IMDb database on multiple metrics. Lower and to the left is better. Note the x-axis log scale.

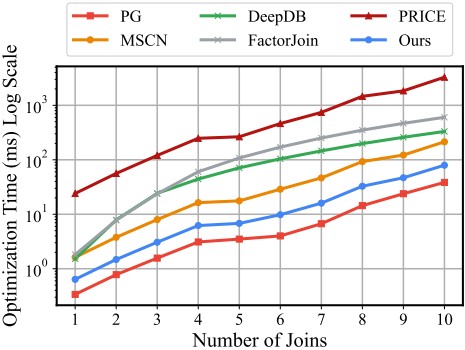

Figure 4: Query Optimization Time Comparison per query on the IMDb dataset. Note the log scale on the Y-Axis.

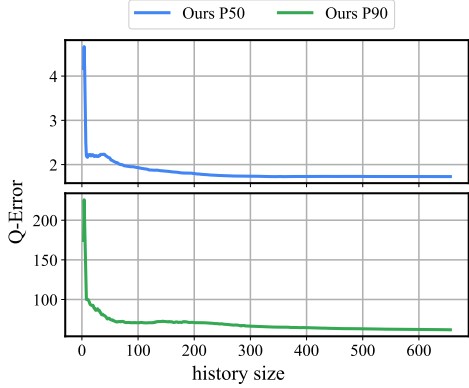

Figure 5: Cumulative Q-Error percentile on the IMDb workload VS size of set $\mathcal{D}_{\mathcal{G}}^{\mathcal{H}}$ (§3.4)

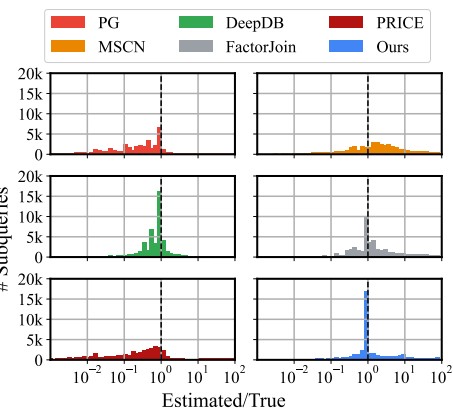

Figure 6: Relative Estimation Errors Histogram on all 46,928 subqueries of IMDb.

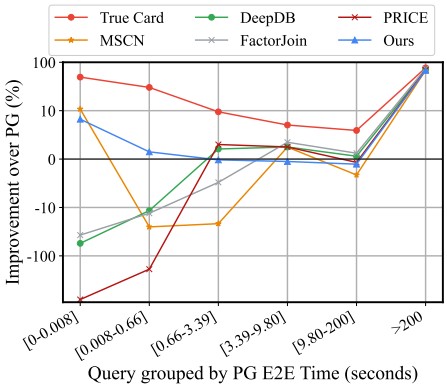

Figure 7: Relative E2E time improvement over PostgreSQL by runtime group. $>0$ means improvements.

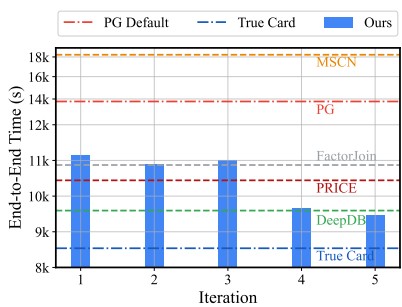

Figure 8: E2E on IMDb. Runtime continuously improves relative to static baselines.

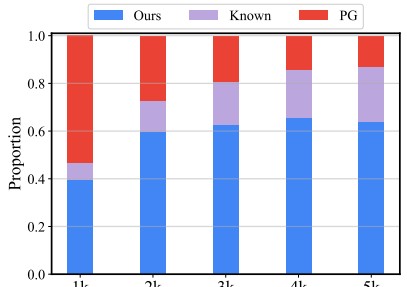

Figure 9: Proportion of reliance on our models VS Postgres as history $\mathcal{D}$ accumulates while simulating the 5k IMDb workload.

Figure 10: **(left)** Subqueries and their cardinalities arrive online, and get stored onto a **(right)** `HashTable` whose entries are keyed by (hash of) graph pattern, and the values are features extracted from graphs matching the pattern. The entries can be arranged as a hierarchy. Inference on test graph $\mathcal{G}$ walks the hierarchy from-right-to-left. If HashTable stores many observations under key $h^{\mathcal{H}_3}(\mathcal{G})$, then the entry's values will be used for inference. If there are only few observations, then the process is repeated with $\mathcal{H}_2, \ldots$, falling-back onto heuristic cost-estimator for novel patterns.

Table 2: Workload Stats. IMDb is from Leis et al. (2015) and others are from Chronis et al. (2024)

| Workload | Tables | Columns | Rows | Join Paths | Queries | Joins | Templates |
|---|---|---|---|---|---|---|---|
| IMDb | 6 | 37 | 62M | 15 | 4972 | 1-4 | 40 |
| stackoverflow | 14 | 187 | 3.0B | 13 | 16,000 | 1-5 | 1440 |
| airline | 19 | 119 | 944.2M | 27 | 20,000 | 1-5 | 1400 |
| accidents | 3 | 43 | 27.4M | 3 | 29,000 | 1-2 | 1450 |
| cms | 24 | 251 | 32.6B | 22 | 14,000 | 1-5 | 2380 |
| geo | 16 | 81 | 8.3B | 15 | 13,000 | 1-5 | 780 |
| employee | 6 | 24 | 28.8M | 5 | 62,000 | 1-5 | 2480 |

estimates should correspond to query plans that execute faster in practice. We replace Posgres' cardinality estimator with alternatives in §4.1, reporting the speedups or slowdowns, while also quantifying the cost required for updating the model or model inference.

- **Standard test/train accuracy**. While our method learns online – with negligible update cost, due to simplicity of chosen regressors, it is also important to compare its metrics with other neural models while restricting to the same train-test data splits. We conduct this comparison in §4.2.

- **Ablation studies**. We want to quantify the importance of using multiple hierarchy levels that are described in §3.5. We ablate different levels of the hierarchy in §4.3.

We report well-established **Q-Error** metric (Moerkotte et al., 2010) that quantifies the ratio of the predicted ($\hat{y}$) from the true cardinality ($y$). Lower is better, with 1 implying perfect estimation.

$$Q_{\text{err}} = \max\left(y/\hat{y},\ \hat{y}/y\right) \tag{13}$$

To understand both typical and tail estimation errors, we report Q-errors at percentiles $\{50, 90, 95\}$.

Further, and more importantly for the user, we report the following run times: **End-to-End (E2E)** query-to-response latency, measured by replacing cardinality estimation of PostgreSQL (v 13.1) with (aforementioned) alternative techniques, per work of Han et al. (2021); **Optimization time** spent by the query optimizer to generate a plan, including the time to obtain cardinality estimates for all subqueries considered by the optimizer; **Overhead time** required for training or updating the cardinality estimation model. For offline, data-driven or query-driven approaches, this is bulk training time. For our online approach, this is the time for incremental updates. Note: we **do not** include the significant overhead of training data collecting for query-driven methods, *e.g.*, $\approx 34$ hours for MSCN.

Table 3: Time-accuracy tradeoffs for IMDb 5K. For every chosen model (left), we report runtime metrics (right) – total execution time (over all 5K queries) and *mean latency* added due to cardinality estimation (that is called a variable number of times, during query planning); and estimation $Q_{err}$ at percentiles (50, 90, 95).

| Model (Category) | Response time | | Time to | Q-Error @ | | |
| | Execution | Latency | tune | P50 | P90 | P95 |
|---|---|---|---|---|---|---|
| POSTGRESQL | 18.9hr | 1.3ms | 4.2s | 4.63 | 193.00 | 948.15 |
| ORACLE | 11.2hr | / | / | 1.00 | 1.00 | 1.00 |
| MSCN (Query-driven) | 24.8hr | 5.4ms | 24m | 4.07 | 70.39 | 219.31 |
| DEEPDB (Data-driven) | 12.6hr | 20ms | 1.5hr | 1.41 | 5.31 | 11.98 |
| FACTORJOIN (Data-driven) | 14.7hr | 20ms | 1.5hr | 2.08 | 34.26 | 92.99 |
| PRICE | 13.4hr | 76ms | 45s | 5.23 | 197.27 | 517.31 |
| PRICE (FT on Queries) | 13.8hr | 76ms | 4hr | 5.02 | 73.69 | 117.41 |
| LITECARD (Online-learn) | 13.9hr | 2.4ms | 37s | 1.70 | 77.12 | 350.19 |

## 4.1 RUN-TIME EXPERIMENTS ON IMDB 5K

Achieving high estimation accuracy often comes at the cost of increased computation, creating a trade-off between accuracy (estimation and lower E2E time) and overheads (model updates and inference). Practical estimator should reside on the Pareto frontier in this multi-dimensional space.

This section uses the IMDb dataset (Leis et al., 2015) for twofold. It was developed to measure execution runtime, ranging from simple to complex queries; Further, all considered SoTA baselines are able to process all queries presented in IMDb workload patterns – even though they cannot process string-predicates or disjunctions out of the box. **Techniques.** We compare LITECARD against default POSTGRESQL and representative state-of-the-art learned estimators across different paradigms: workload-driven (MSCN), data-driven (DEEPDB, FACTORJOIN), and zero-shot (PRICE). **Hardware.** All experiments were conducted on a 64-Core AMD EPYC 7B13 CPU and 120GB RAM. Like Han et al. (2021), we ran POSTGRESQL on a single CPU and disabled GEQO[2]. We instantiate our LITECARD implementation with an empty hierarchy, and divisively populate it for every executed subquery. For inference, (§3.4 & §3.5), we set $\beta_3 = 10$, $\beta_2 = 50$, $\beta_1 = 100$.

**Overall Performance and Efficiency Comparison.** Table 3 and Figure 3 compares performance (End-to-End Time, Q-Error) and cost (Optimization Time, Training Time) across all techniques on the 5k IMDb workload. We make the following obervations.

- Executing all 5K queries of IMDb workload on unmodified Postgres takes about **18.7 hours**. To establish a lower-bound on run-time, we replace Postgres' cardinality estimates by an Oracle, allowing the plan optimizer to recover the optimal query plans. The lower-bound is **11.2 hours**.
- On this dataset, the best cardinality estimates come from DEEPDB, and it gives the least execution time. However, tuning takes ∼1.5 hours on the table contents, which must be done separately for every database, and potentially repeated as database contents shift. Further, the latency of all models is considerably higher than LITECARD.
- Latency is the time it takes for the DB engine to start piping records on the response stream, i.e., after the query planner had produced query plan, which involves invoking the cardinality estimator many times (up-to thousands of times). Since LITECARD utilizes simple models (hashing, followed by simple regressor), the inference time and therefore query planning time, is significantly lower than neural methods.
- LITECARD substantial improves on Postgres, with 1.36X speedup in total Execution Time (13.9hr vs 18.9hr) and better Q-errors (1.70 vs 4.63 at P50, and 77.12 vs 193.00 at P90). Crucially, it does this while **maintaining small latency**, compared to other alternatives. It also incurs a negligible training overhead (37.3s total for the 5k query workload) than any other learned method. The tuning time includes: hashing (Eq. 11) and feature extraction (Eq. 12) together averaging $30\mu s$ per subquery, as well as fitting a decision forest, averaging $700\mu s$ per subquery.

**Optimization Time Scalability.** Figure 4 shows that cardinality estimation time **scales exponentially** with query complexity (number of joins). Therefore, practical cardinality estimators must

---

[2]https://github.com/Nathaniel-Han/End-to-End-CardEst-Benchmark

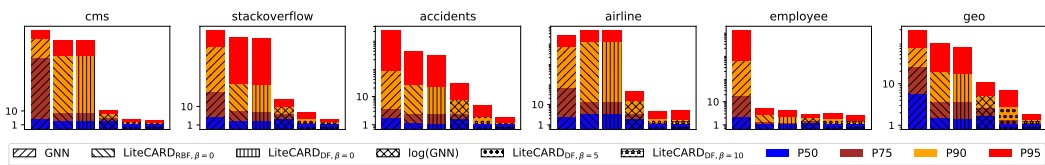

Figure 11: Test Q-errors with train:test as 50:50 (disabling online-learning for LITECARD).

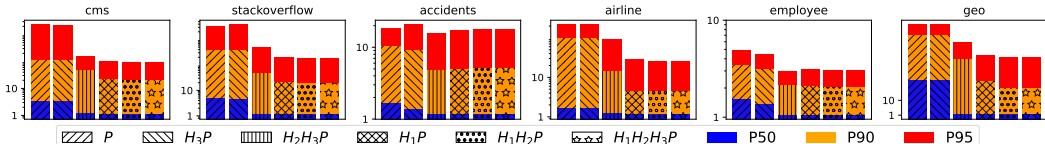

Figure 12: Ablating the depth of the hierarchy depicted in Fig. 10.

exhibit minimal latency. The figure shows that default POSTGRESQL starts with low optimization time ($\approx 0.3$ ms for 1 join) and increases gradually. LITECARD mirrors this behavior, remaining comparable to POSTGRESQL across all join counts (*e.g.*, $\approx 60 - 80$ ms at 10 joins), which is feasible because our lightweight models enable per-subquery estimates in $\approx 0.1$ ms. In contrast, other baslines slow optimization by 10X-100X, posing a major practical barrier.

## 4.2 STANDARD TRAIN:TEST SPLITS

While online-learning is important for lowering practical barriers plug-and-play, it is also important to compare accuracy of LITECARD compared to other baselines, while having access to exactly the same training data without allowing (online) incremental learning from the test data. We evaluate this setting on datasets from CardBench (Chronis et al., 2024) (bottom-half of Table 2). The queries contain a variety of conjunctions, disjunctions, string predicates, and up to 5 joins. Unfortunately, many baselines lack support for these complexities, *e.g.*, DeepDB, MSCN, PRICE lack string predicates and disjunction support.

We download all queries from CardBench, and partition each workload into train:test with ratio 50:50. We fit Graph Neural Network (architecture in Appendix C) on the training split Fig. 11 uses only $\mathcal{H}_3$ against . We observe that our models become accurate when $\beta \geqslant 5$ – i.e., suggesting we should trust it for subquery patterns that repeat just a few times.

For evaluating LITECARD, each test inference had access only to training data, without any other test examples. This disables the online-learning capability of our contribution for these experiments.

## 4.3 ABLATION STUDIES

We compare various levels of the hierarchy depicted in Fig. 10 (§3.5), including one-level hierarchy – consisting of only the traditional cardinality estimator (Postgres), the full hierarchy ($\mathcal{H}_3 \rightarrow \mathcal{H}_2 \rightarrow \mathcal{H}_1 \rightarrow$ P), and other in-between options. As summarized in Fig. 4.3, deeper hierarchies generally show better performance at various Q-error percentiles. Details are in Appendix G.1, Table 5.

## 5 RELATED WORK

We review a variety of cardinality estimation methods. **Traditional** techniques (PostgreSQL Group, 2025; OracleMySQL, 2024; Lipton et al., 1990; Leis et al., 2017), such as histogram-based methods and sampling-based approaches, rely on simplified assumptions about data distributions and attribute independence. While efficient and easily updatable, they often struggle with complex query patterns involving multiple joins, and correlated data, leading to large estimation errors. In recent years, several lines of learned cardinality estimation have been proposed (Han et al., 2021; Sun et al., 2021; Kim et al., 2022). These approaches can be broadly grouped into query-driven, data-driven, and zero-shot. **Query-driven methods** frame cardinality estimation as a supervised learning problem,

training models to map featurized query to cardinality – *e.g.*, feed-forward networks (Kipf et al., 2019; Reiner & Grossniklaus, 2024), gradient boosted trees (Dutt et al., 2019), and tree-LSTM (Sun & Li, 2019). These methods require training data **upfront** (rather than online) *i.e.*, simulating and executing queries while recording their cardinalities. Training may be repeated when database contents or workloads shift. Further, they add an overhead during query planning (inference) (§4.1). Our method is also supervised, though learns many simple models, online, one model per subquery pattern. Pattern-based learning has appeared before: for example, (Malik et al., 2007) group queries by pattern, and perform learning-and-inference on dense-vectors within each pattern; Woltmann et al. (2019) learns local models for queries that share same tables; Dutt et al. (2019) creates conjunction trees from simple predicates. However, we differ in: (1) our patterns are graph rather than SQL text, which are invariant to aliases and ordering (e.g., of junctions); and (2) learning hierarchy of models rather than a one-level partitioning. **Data-driven Methods** directly model the table data distributions (Tzoumas et al., 2011; Hilprecht et al., 2020; Yang et al., 2019; 2021; Zhu et al., 2021; Wu et al., 2023; Kim et al., 2024). They generally produces effective estimates and results in good end-to-end time performance. However, they typically incur long training time, large model size and slow optimization time. Updating these models when the underlying data changes is also slow and often requires expensive re-training. **Zero-shot Methods** aim to transfer knowledge learned from a diverse set of pre-trained databases to a new database without requiring database-specific training data Zeng et al. (2024). While promising for cold-start scenarios, these methods can still suffer from high optimization time. Furthermore, while they can be fine-tuned on database-specific queries, this process can still be slow. **Other query optimization techniques.** Flow-Loss (Negi et al., 2020; 2021) trains learned cardinality estimators with a plan-aware loss that treats different cardinality errors differently based on their impact on query plans, instead of relying on generic metrics like Q-error.

## 6 CONCLUSION

We are interested in learning a cardinality estimator for diverse workloads. Instead of a monolithic model that can handle any arbitrary query, we learn many simple models, each model specialized to one subquery pattern. In particular, we define cardinality estimation models using a kernel function across Graphs. The kernel deems two subqueries as similar if they are structurally-equivalent and they have similar features. Similar subqueries influence one another either when learning a local model (Eq. 5) or with one-shot inference (Eq. 7). We presented an efficient implementation using an online learning algorithm that extracts (feature-vector, cardinality) pair for every subquery graph, and groups them by graph hash values. Finally, we configure multiple hash functions and their corresponding learning features, such that, the query history can be recursively partitioned into a hierarchy. The leaves of the hierarchy contain subqueries that are highly-similar (*e.g.*, equivalent, up-to constants and literals), whereas first and intermediate levels of the hierarchy aggregate more general queries, where nodes contain structurally-equivalent subqueries that read different columns or use different op-codes. Our method provides a uniquely compelling balance, achieving significant performance benefits and accuracy improvements over traditional methods with operational costs orders of magnitude lower than other learned techniques, positioning itself on the practical Pareto frontier for learned cardinality estimation.

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

APPENDIX

## A  DIRECTED ACYCLIC GRAPHS OF SQL QUERIES

We convert an input SQL query ([3]) into a directed acyclic graph (DAG) in the following steps:

1. Parse input statement as a parse-tree. It is possible to use an open-source parser, like `https://github.com/tobymao/sqlglot`.
2. Merge identical nodes (column names or table names).
3. For every referenced *column*, we add two edges: Table → Table Alias[4] → *column*.

The parse-tree (Step 1 above) already contains the predicate expression tree appearing in the "WHERE"-clause, *e.g.*, with nodes representing column names; operators (=, >, +, not, ...); conjuctions and disjunctions (and, or); literals; function names (SUBSTRING, ABS, NOW, ...); etc.

## B  BASELINES

- POSTGRESQL (PostgreSQL Group, 2025). Denotes POSTGRESQL's cardinality estimator.
- ORACLE. Emits the correct cardinality, establishing lower-bounds on errors and runtimes.
- MSCN (Kipf et al., 2019): Multiset neural network that learns: query → cardinality. The model was trained using author-provided code for 200 epochs.
- DEEPDB (Hilprecht et al., 2020): data-driven approach that learns a sum-product network for each selected subset of tables in the database.
- FACTORJOIN (Wu et al., 2023): a data-driven approach that applies factor graph on single tables and aggregates histograms for multiple tables.
- PRICE (Zeng et al., 2024): zero-shot approach, with parameters pre-trained on 30 datasets. The overhead time for the base zero-shot model (45s in Table 3) is incurred for computing necessary statistics such as histograms, fanout, common value counts, and table sizes.
- PRICE (FT) We fine-tuned the above, using their code-base, on 50k queries for 100 epochs.

## C  GNN BASELINE

We first convert the heterogeneous graph into a **latent homogeneous graph**, then develop a 4-layer GNN with residual connections.

In particular, we train one shallow network **per node type** that maps the node's features onto fixed $d$-dimensional space which we denote here by $\mathbf{Z} \in \mathbb{R}^{n \times d}$. Let $\mathbf{A}$ denote the adjacency matrix.

The GNN outputs $\widehat{y}$, calculated as:

$$\mathbf{Z}^{(0)} \triangleq \mathbf{Z} \tag{14}$$

$$\mathbf{Z}^{(\ell+1)} = \mathbf{Z}^{(\ell)} + \text{MLP}_\ell \left( \text{concat} \left( (\widetilde{\mathbf{A} + \mathbf{I}})\mathbf{Z}^{(\ell)}, (\widetilde{\mathbf{A}^\top + \mathbf{I}})\mathbf{Z}^{(\ell)} \right) \right) \mathbf{W} \tag{15}$$

$$\widehat{y} = \text{Readout}(\mathbf{Z}^{(4)}) \tag{16}$$

where $\widetilde{\ }$ divides each entry by the sum of its row, *i.e.*, $\widetilde{M} = M \times \text{diag}(M \times \mathbf{1})^{-1}$, and $\mathbf{I}$ is identity matrix. We use ReLu activation for the multi-layer perceptrons (MLPs).

We train two variants, one with R.M.S.E objective *i.e.*, $\min \|y - \widehat{y}\|$, and anotherwith R.M.S.log.E, *i.e.* with, $\min \| \log(y) - \log(\widehat{y})\|$. Respectively referred to as, GNN and log(GNN), in the manuscript.

---

[3]See Appendix for PostgreSQL's RelInfo data structure

[4]The alias is important as certain queries access one table twice, joining it with itself. Nonetheless, the alias name is ignored by our method.

# D INTEGRATION WITH POSTGRESQL

To evaluate the efficacy of LITECARD, we integrated it into open-source PostgreSQL as an extension, as depicted in Figure 13. This integration involved adding new hooks into the PostgreSQL engine, enabling the query planner to utilize LITECARD for cardinality estimation, thereby influencing plan decisions and allowing the collection of performance statistics to demonstrate the efficacy of LITECARD approach. While this work focuses on demonstrating the core algorithm's efficacy, production-level optimizations such as memory management, storage and asynchronous training mechanisms are are beyond the scope of this paper.

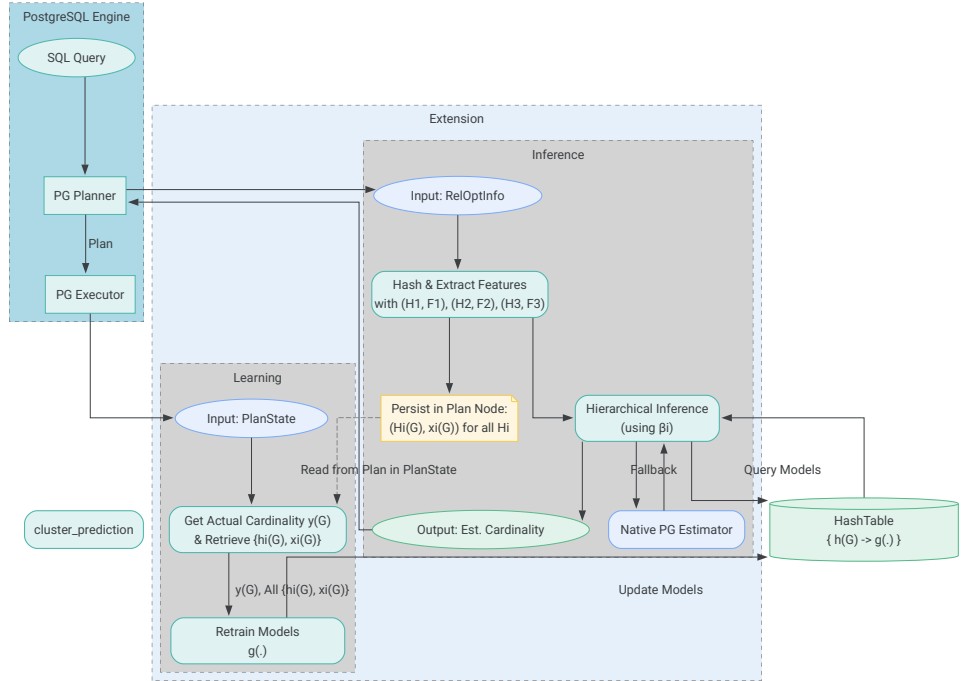

Figure 13: Integrating LITECARD with PostgresSQL

## D.1 INFERENCE

LITECARD interacts with the cost estimator at various points within the PostgreSQL planner to provide learned estimates. This is achieved using PostgreSQL's hook mechanism, specifically by setting hooks within functions such as `set_baserel_size_estimates` (PG cardinality estimation function for base relations) and `get_parameterized_joinrel_size` (PG cardinality estimation function for join relations) and more. These hooks allow us to override the default cardinality estimates. When the planner requires a cardinality for a relation (represented by `RelOptInfo`), our hooks are invoked. We process the `RelOptInfo` struct, analyzing filters (`baserestrictinfo`), join information, and other plan attributes to generate hashes and corresponding features according to the strategies defined in §3.2. The system attempts to predict cardinality using the model corresponding to $\mathcal{H}_3$. Following the hierarchical approach outlined in §3.5, if the model for $\mathcal{H}_3$ does not meet the activation threshold $\beta_1$ (e.g., insufficient training samples), we fallback to the previous level in the hierarchy, $\mathcal{H}_2$, generating $h^{\mathcal{H}_2}(G)$ and $\mathbf{x}^{\mathcal{H}_2}(G)$ to invoke the corresponding $g(.)$. This process continues to $\mathcal{H}_1$ if necessary. If no model in the hierarchy is sufficiently confident, we fallback to the native PostgreSQL estimator, ensuring robustness. The metadata generated during this process, including the hashes $\left(h^{\mathcal{H}_1}(G), h^{\mathcal{H}_2}(G), h^{\mathcal{H}_3}(G)\right)$ and the extracted features $\left(\mathbf{x}^{\mathcal{H}_1}(G), \mathbf{x}^{\mathcal{H}_2}(G), \mathbf{x}^{\mathcal{H}_3}(G)\right)$, and which hierarchical level provided the estimate, are persisted within the plan node structures (specifically within the Plan nodes). This information is crucial for online learning and observability.

Table 4: PG (Biased) Cardinality Estimation Analysis on the IMDb database. Note that as the number of joins increases, the underestimate proportion and average Q-error increase drastically.

| $n\_join$ | Underestimate Proportion | Average Q-Error |
|---|---|---|
| 1 | 0.57 | 1.57 |
| 2 | 0.83 | 20.20 |
| 3 | 0.93 | 1361.38 |
| 4 | 0.98 | 68655.97 |

## D.2 LEARNING

The online learning mechanism (§3) is realized through executor hooks. We use the `ExecutorStart_hook` to ensure row count instrumentation is enabled for each node in the plan. The `ExecutorEnd_hook` is pivotal for capturing the ground truth after query execution. Once execution is complete, for each node in the plan tree, we retrieve the persisted hash value $h^{\mathcal{H}_i}(\mathcal{G})$ and features $\mathbf{x}^{\mathcal{H}_i}(\mathcal{G})$, along with the actual cardinality $y$ from the execution statistics. This triplet $\left(h^{\mathcal{H}_i}(\mathcal{G}), \mathbf{x}^{\mathcal{H}_i}(\mathcal{G}), y\right)$ constitutes a new training example. This example is used to update or retrain the parameters of the corresponding model $g(.)$, thus allowing the models to continuously adapt to the observed query workload.

## D.3 OBSERVABILITY

To facilitate understanding of LITECARD's behavior, we have enhanced the EXPLAIN ANALYZE command of PostgreSQL. The output for each plan node now includes the cardinality predicted by LITECARD, the inference latency for the LITECARD model, the hash $h^{\mathcal{H}_i}(\mathcal{G})$ used for the prediction, the features $\mathbf{x}^{\mathcal{H}_i}(\mathcal{G})$ extracted and the hierarchical level $i$ from which the prediction was made.

## D.4 HANDLING POSTGRESQL BIAS

Effectively integrating a learned estimator requires understanding and mitigating biases in the base optimizer. PostgreSQL's default estimator exhibits a significant underestimation bias, which can impede optimal plan selection.

**POSTGRESQL's Underestimate Bias.** Table 4 quantifies the inherent underestimation bias in PostgreSQL's default cardinality estimates on the IMDb JOB-Light workload (Leis et al., 2015). The table shows the proportion of subqueries underestimated by PostgreSQL and their average Q-error, grouped by join count. We observe the underestimation proportion sharply increases with joins (e.g., >80% for 2-join, >98% for 4-join queries). Correspondingly, average Q-error escalates dramatically, reaching over 68,000 for 4-join queries. This systematic underestimation is critical as optimizers rely on these estimates for plan choices; underestimates can lead PostgreSQL to select seemingly cheaper but suboptimal plans (e.g., favoring nested loops for intermediate results that are much larger than estimated). Table 4 demonstrates PostgreSQL's severe, join-dependent underestimation bias, a key factor leading to poor plan quality.

**Impact of Bias and Our Solution.** Figure 14 illustrates the impact of POSTGRESQL's bias using an example query from the 5000-query IMDb workload. If we naively combine estimates, POSTGRESQL's underestimate for subqueries lacking historical data (represented by the red nodes) leads to a disastrous plan executing in 3400 seconds. This occurs because POSTGRESQL's underestimate makes these subqueries appear smallest at their level, causing the optimizer to select them. To address this severe underestimate bias problem, we sample a probability number and then multiply their POSTGRESQL estimates by the average Q-errors documented in Table 4. For example, for a subquery at the third level involving 2 joins, we uniform sample a probability from 0 to 1, if it is smaller than 0.83 , we multiply the estimate by 20.2; for a fourth-level subquery involving 3 joins, if the sampled number is smaller than 0.98, we multiply by 1361.38. This bias information (*e.g.* Table 4) can be practically collected from executed queries for any database with minimal overhead. Figure 14 shows that applying this adjustment allows LITECARD to avoid the disastrous plan,

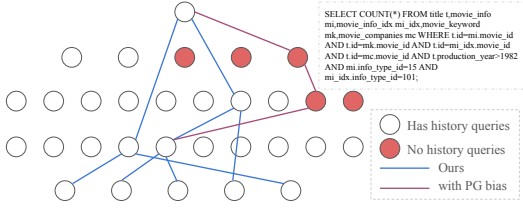

Figure 14: Query planning example illustrating the impact of PostgreSQL bias. Each node represents a subquery where the bottom level are the single table queries and the top node is the whole query. Shows how an underestimate can lead to a disastrous plan path (3400s execution) and how adjusting the bias allows LITECARD to select a better plan (141s execution).

resulting in a near-optimal execution time of 141 seconds, compared to PostgreSQL's default plan at 171 seconds and injecting true cardinality oracle at 133 seconds.

## E  CORRECTNESS PROOFS

**Definition 1.** *(Graph Isomorphism under feature set) Let graphs $\mathcal{G}$ and $\mathcal{G}'$ be isomorphic under feature-set $\mathcal{H}$, denoted as* $\boxed{\mathcal{G} \underset{\mathcal{H}}{\cong} \mathcal{G}'}$ *if-and-only-if there exists a bijection $\pi_{(.)} : \mathcal{V} \to \mathcal{V}'$ such that*

$$\mathcal{E}' = \{(\pi_u, \pi_v)\}_{(u,v)\in\mathcal{E}} \quad \textbf{and} \quad \mathcal{X}'_{\pi_j}[(t,a)] = \mathcal{X}_j[(t,a)] \, \textit{for all} \, (t,a) \in \mathcal{H} \, \textit{and} \, j \in \mathcal{V} \qquad (17)$$

**Definition 2.** *(Predecessors) Let $\mathcal{P}_j \subset \mathcal{V}$ be the predecessors to node $j \in \mathcal{V}$ defined as follows. Given edge $(u,v) \in \mathcal{E}$, its starting-point $u$ will be included in $\mathcal{P}_j$ if either $v = j$ or $v \in \mathcal{P}_j$.*

**Definition 3.** *(Successors) Let $\mathcal{S}$ the equals the $\mathcal{P}$ corresponding to the reverse graph $(\mathcal{V}, \mathcal{E}^\top, \mathcal{X})$.*

**Theorem 1.** *Any feature set $\mathcal{H} \subseteq \mathcal{A}$ can induce a canonical node ordering. Specifically,*

$$\mathcal{G} \underset{\mathcal{H}}{\cong} \mathcal{G}' \implies \mathbf{A}^{\pi^{\mathcal{H}}(\mathcal{G})} \times \mathbf{H}^{\mathcal{H}}(\mathcal{G}) = \mathbf{A}^{\pi^{\mathcal{H}}(\mathcal{G}')} \times \mathbf{H}^{\mathcal{H}}(\mathcal{G}') \qquad (18)$$

$$\mathcal{G} \underset{\mathcal{H}}{\cong} \mathcal{G}' \underset{whp}{\impliedby} \mathbf{A}^{\pi^{\mathcal{H}}(\mathcal{G})} \times \mathbf{H}^{\mathcal{H}}(\mathcal{G}) = \mathbf{A}^{\pi^{\mathcal{H}}(\mathcal{G}')} \times \mathbf{H}^{\mathcal{H}}(\mathcal{G}'), \qquad (19)$$

*such that $\pi^{\mathcal{H}}(\mathcal{G})$ and $\pi^{\mathcal{H}}(\mathcal{G}')$ can be used to align the featured DAGs, and sparse re-ordering (adjacency) matrix $\mathbf{A}^{\pi^{\mathcal{H}}(\mathcal{G})} \in \{0,1\}^{n \times n}$ shuffles rows of its multiplicand according to ordering defined by $\pi^{\mathcal{H}}(\mathcal{G})$, as:*

$$A_{i,j}^{\pi^{\mathcal{H}}(\mathcal{G})} = \mathbf{1}_{\left[j \, = \, \pi_i^{\mathcal{H}}(\mathcal{G})\right]} \qquad (20)$$

**Proof of Theorem 1.**  We start with implication (Eq. 18), as it is easier to show. Assume that $\mathcal{G}$ and $\mathcal{G}'$ are isomorphic under $\mathcal{H}$. Two graphs $(\mathcal{G}, \mathcal{G}')$ can be isomorphic only if they have the same number of nodes. Let $n = |\mathcal{V}| = |\mathcal{V}'|$. We first show that, in-between and after calculating equations 8 then 9 then 10, the following **property** is maintained: matrices $\mathbf{H}^{\mathcal{H}}$ and $\mathbf{H}'^{\mathcal{H}}$ contain the same rows, but not necessarily in the same order. Then, we show that left-multiplication with $\mathbf{A}$ sorts rows with matching orders.

- Since $(\mathcal{G}, \mathcal{G}')$ are assumed isomorphic under $\mathcal{H}$, therefore $\mathcal{X}$ is just a re-ordering of $\mathcal{X}'$ (per Definition 1. Since $\mathbf{H}_j = \$(\mathcal{X}_j)$ and $\mathbf{H}'_j = \$(\mathcal{X}'_j)$, then $\mathbf{H}$ is just a re-ordering of $\mathbf{H}'$ and therefore the property is maintained after Eq. 8.

- To prove the property is maintained after calculating Eq. 9 follows. TOPOLOGICALORDER processes every node exactly once. Starting from nodes $j$ where $|\mathcal{P}_j| = 0$, the update $\mathbf{H}_j^{\mathcal{H}} := \$\left(\mathbf{H}_j^{\mathcal{H}} \oplus \mathtt{sort}(\{\mathbf{H}_k^{\mathcal{H}} \mid (k,j) \in \mathcal{E}\})\right)$ reduces to $\mathbf{H}_j^{\mathcal{H}} := \$\left(\mathbf{H}_j^{\mathcal{H}}\right)$. More generally, after computing Eq.9 for any $j$, TOPOLOGICALORDER guarantees that the row $\mathbf{H}_j^{\mathcal{H}}$ is exactly a function of $\mathcal{P}_j$ (when restricting to features in $\mathcal{H}$).

- The proof that property is maintained after calculating Eq. 10 mirrors the above but following reverse-topological order of $\mathcal{S}$ in lieu of $\mathcal{P}$.

Finally, the multiplication $\mathbf{A} \times \mathbf{H}$ only re-orders the nodes of $\mathbf{H}$ (per Eq. 20), exactly to sort the rows of $\mathbf{H}$ lexicographically (per Eq. 11). This applies to both $\mathbf{H}^{\mathcal{H}}(\mathcal{G})$ and $\mathbf{H}^{\mathcal{H}}(\mathcal{G}')$.

$$\text{Therefore,} \qquad \mathcal{G} \underset{\mathcal{H}}{\cong} \mathcal{G}' \implies \mathbf{A}^{\pi^{\mathcal{H}}(\mathcal{G})} \times \mathbf{H}^{\mathcal{H}}(\mathcal{G}) = \mathbf{A}^{\pi^{\mathcal{H}}(\mathcal{G}')} \times \mathbf{H}^{\mathcal{H}}(\mathcal{G}').$$

We prove the reverse implication (Eq. 19) by contradiction.

$$\text{For the sake of contradiction, assume:} \quad \mathbf{A}^{\pi^{\mathcal{H}}(\mathcal{G})} \times \mathbf{H}^{\mathcal{H}}(\mathcal{G}) = \mathbf{A}^{\pi^{\mathcal{H}}(\mathcal{G}')} \times \mathbf{H}^{\mathcal{H}}(\mathcal{G}'), \qquad (21)$$

$$\text{and not:} \quad \mathcal{G} \underset{\mathcal{H}}{\cong} \mathcal{G}'. \tag{22}$$

The assumption (Eq. 21) implies that every for any row $j \in \mathcal{V}$, the string (bit vector) $\mathbf{H}_j^{\mathcal{H}}(\mathcal{G}) \in \{0,1\}^{256}$ exists at some row in $\mathbf{H}^{\mathcal{H}}(\mathcal{G}')$. We now show that $\mathbf{H}_j^{\mathcal{H}}(\mathcal{G})$ is a deterministic uniform-random function of $\{\mathcal{X}_k[(t,a)] \mid k \in \{j\} \cup \mathcal{P}_i \cup \mathcal{S}_i\}_{(t,a) \in \mathcal{H}}$, plus the edge structure of $\{j\} \cup \mathcal{P}_i \cup \mathcal{S}_i$ that is linking these feature nodes. Crucially, a bijective function, with high probability (*whp*).

When calculating $\mathbf{H}^{\mathcal{H}}(G)$, each row $\mathbf{H}_j^{\mathcal{H}}$ will be updated once in each of Equations 8, 9, and 10, *i.e.*, thrice. First updates (Eq.8) can happen to all nodes in-parallel. Second updates (Eq.9) happen in topological order, and third updates happen in reverse-topological order (Eq.10).

- After first set of updates (Eq. 8), $\mathbf{H}_j^{\mathcal{H}} = \$\left(\oplus\{\mathcal{X}_j[(t,a)]\}_{(t,a) \in \mathcal{H}}\right)$ encorporate into $\mathbf{H}_j$ the features of nodes $\{j\}$.
- The second set of updates proceeds in topological order. For leaf nodes, they will just re-hash their their features *i.e.* $\mathbf{H}_j = \$\left(\$\left(\{\mathcal{X}_j[(t,a)]\}\right)\right)$. Subsequent (non-leaf node) node $j$ updates its hash, by concatenating the current $\mathbf{H}_j$ (already capturing $\mathcal{X}_j$), with already updated hashes of their incoming neighbors $\{\mathbf{H}_k\}_{(k,j) \in \mathcal{E}}$. This update includes the in-degree *local structure*. Since each neighbor $\mathbf{H}_k$ has already updated from its predecessor neighbors, then recursively and by induction, each node $j$ updates its hash to a deterministic function of features of all nodes $\in \{j\} \cup \mathcal{P}_j$.
- Echoing the above, but in reverse topological order, updates string $\mathbf{H}_i$ to its final value, a deterministic function of features of nodes all nodes $\in \{j\} \cup \mathcal{P}_j \cup \mathcal{S}_j$.

It is important to realize that hashing function $\$(.)$ is run on its own output (like $\$(\$(.))$). We wish to have the output to be uniform – *i.e.*, each outcome has $\approx \frac{1}{2^{256}}$ to appear. We are therefore restricted to cryptographic hashing functions. In practice, we use MD5. This shows that:

$$\mathbf{A}^{\pi^{\mathcal{H}}(\mathcal{G})} \times \mathbf{H}^{\mathcal{H}}(\mathcal{G}) = \mathbf{A}^{\pi^{\mathcal{H}}(\mathcal{G}')} \times \mathbf{H}^{\mathcal{H}}(\mathcal{G}') \underset{whp}{\implies} \mathcal{G} \underset{\mathcal{H}}{\cong} \mathcal{G}' \tag{23}$$

$\square$

**Theorem 2.** *The sets $\mathcal{H} \subseteq \mathcal{A}$ and $\mathcal{H} \subseteq \mathcal{A}$ can extract a canonical feature vector. Specifically,*

$$\mathcal{G} \underset{(\mathcal{H} \cup \mathcal{F})}{\cong} \mathcal{G}' \implies \mathbf{x}_{\mathcal{F}}^{\mathcal{H}}(\mathcal{G}) = \mathbf{x}_{\mathcal{F}}^{\mathcal{H}}(\mathcal{G}') \tag{24}$$

**Proof of Theorem 2.** We copy Eq. 12:

$$\mathbf{x}_{\mathcal{F}}^{\mathcal{H}} = \bigoplus_{j \in \pi^{\mathcal{H}}} \left\{ f_{(t,a)}(\mathcal{X}_j[(t,a)]) \mid t = \tau_j \right\}_{(t,a) \in \mathcal{F}}$$

which rasterizes node features into a flat vector, using the ordering dictated by $\pi^{\mathcal{H}}(G)$. We are given that: $\mathcal{G} \underset{(\mathcal{H} \cup \mathcal{F})}{\cong} \mathcal{G}'$. But,

$$\mathcal{G} \underset{(\mathcal{H} \cup \mathcal{F})}{\cong} \mathcal{G}' \implies \mathcal{G} \underset{\mathcal{H}}{\cong} \mathcal{G}'$$

as the right-side is less restrictive. Using Theorem1, $\pi^{\mathcal{H}}(\mathcal{G})$ corresponds to $\pi^{\mathcal{H}}(\mathcal{G}')$, specifically equating

$$\bigotimes_{j \in \pi^{\mathcal{H}}(\mathcal{G})} \{\psi(\mathcal{X}_j)\} = \bigotimes_{j \in \pi^{\mathcal{H}}(\mathcal{G}')} \{\psi(\mathcal{X}_j')\} \tag{25}$$

for any arbitrary function $\psi(.)$ and any (ordered set) aggregation function $\otimes$. Choosing $\otimes$ as $= \oplus$ and $\psi(.) = \left\{ f_{(t,a)}(.[(t,a)]) \mid t = \tau_j \right\}_{(t,a)\in\mathcal{F}}$ recovers that $\mathbf{x}_{\mathcal{F}}^{\mathcal{H}}(\mathcal{G}) = \mathbf{x}_{\mathcal{F}}^{\mathcal{H}}(\mathcal{G}')$. Therefore,

$$\mathcal{G} \underset{(\mathcal{H}\cup\mathcal{F})}{\cong} \mathcal{G}' \implies \mathbf{x}_{\mathcal{F}}^{\mathcal{H}}(\mathcal{G}) = \mathbf{x}_{\mathcal{F}}^{\mathcal{H}}(\mathcal{G}')$$

$\square$

**Theorem 3.** *Given an arbitrary anchor graph $\mathcal{G}$, then every $\mathbf{x} \in \{\mathbf{x}_{\mathcal{F}}^{\mathcal{H}}(\mathcal{G}') \mid h(\mathcal{G}) = h(\mathcal{G}')\}$ has the same dimensionality, with canonical node-to-feature positions.*

**Proof of Theorem 3** From Theorem 1, we have:

$$\mathbf{A}^{\pi^{\mathcal{H}}(\mathcal{G})} \times \mathbf{H}^{\mathcal{H}}(\mathcal{G}) = \mathbf{A}^{\pi^{\mathcal{H}}(\mathcal{G}')} \times \mathbf{H}^{\mathcal{H}}(\mathcal{G}') \underset{whp}{\implies} \mathcal{G} \underset{\mathcal{H}}{\cong} \mathcal{G}'$$

Moreover, we have that:

$$\mathbf{A}^{\pi^{\mathcal{H}}(\mathcal{G})} \times \mathbf{H}^{\mathcal{H}}(\mathcal{G}) = \mathbf{A}^{\pi^{\mathcal{H}}(\mathcal{G}')} \times \mathbf{H}^{\mathcal{H}}(\mathcal{G}') \implies h^{\mathcal{H}}(\mathcal{G}) = h^{\mathcal{H}}(\mathcal{G}'), \tag{26}$$

which follows from the definition of $h^{\mathcal{H}}(.)$ in Eq. 11 as:

$$h^{\mathcal{H}}(\mathcal{G}) = \$\left( \bigoplus_{j\in\pi^{\mathcal{H}}} \mathbf{H}_j^{\mathcal{H}}(\mathcal{G}) \right) = \$\left( \bigoplus_{j\in\{1,2,\dots,n\}} \left[ \mathbf{A}^{\pi^{\mathcal{H}}(\mathcal{G})} \times \mathbf{H}^{\mathcal{H}}(\mathcal{G}) \right]_j \right)$$

$$= \$\left( \bigoplus_{j\in\{1,2,\dots,n\}} \left[ \mathbf{A}^{\pi^{\mathcal{H}}(\mathcal{G}')} \times \mathbf{H}^{\mathcal{H}}(\mathcal{G}') \right]_j \right) = h^{\mathcal{H}}(\mathcal{G}')$$

The converse of Eq. 26 holds with high probability, specifically, since $\$$ is a uniform hashing function, *i.e.*, producing 1-to-1 mapping (with collision rate of $\frac{1}{2^{256}}$). Therefore, we have:

$$h^{\mathcal{H}}(\mathcal{G}) = h^{\mathcal{H}}(\mathcal{G}') \underset{whp}{\implies} \mathbf{A}^{\pi^{\mathcal{H}}(\mathcal{G})} \times \mathbf{H}^{\mathcal{H}}(\mathcal{G}) = \mathbf{A}^{\pi^{\mathcal{H}}(\mathcal{G}')} \times \mathbf{H}^{\mathcal{H}}(\mathcal{G}')$$

$$\text{hence,} \quad h^{\mathcal{H}}(\mathcal{G}) = h^{\mathcal{H}}(\mathcal{G}') \underset{whp}{\implies} \mathcal{G} \underset{\mathcal{H}}{\cong} \mathcal{G}'.$$

Finally, Theorem 3 considers pairs for which $h(\mathcal{G}) = h(\mathcal{G}')$. Therefore, with high probability (due to above), $\mathcal{G} \underset{\mathcal{H}}{\cong} \mathcal{G}$. Therefore, the ordering $\pi^{\mathcal{H}}(\mathcal{G})$ must be consistent with $\pi^{\mathcal{H}}(\mathcal{G}')$. The sequence of node **types**, when iterating over $\mathcal{G}$ per $\pi^{\mathcal{H}}(\mathcal{G})$, must be the same sequence of node types when iterating over $\mathcal{G}'$ per $\pi^{\mathcal{H}}(\mathcal{G}')$. During these iterations, the vectors $\mathbf{x}_{\mathcal{F}}^{\mathcal{H}}(\mathcal{G})$ and $\mathbf{x}_{\mathcal{F}}^{\mathcal{H}}(\mathcal{G}')$ are composed. Since the feature dimension is deterministic given a node type, then (each type, structural position) will occupy distinct positions in the feature vectors. $\square$

As an aside, in our implementation, we also always include these features for all nodes: in-degree, out-degree, and node type (table, column, operand, ...) and always include them in $\mathcal{H}$.

## F Feature Extractors

We define several functions. Each can extract node features. For any node, its entire feature vector is the concatenation of all applicable feature extractors. We implement a handful of $f$'s:

- ($f_1$) $f_{\text{num}}(m) = m \in \mathbb{R}^1$. Applies to numeric literals. Casting from string to number is implied.

- ($f_2$) $f_{\text{scaled}}(m) = \frac{m-\text{minVal}(\mathbf{c})}{\text{maxVal}(\mathbf{c})-\text{minVal}(\mathbf{c})} \in \mathbb{R}^1$. Applies to numeric literals when used alongside column $\mathbf{c}$. It can be activated if the DB engine stores min- and max-value per column.

- ($f_3$) $f_{\text{comp}}(m) \in \mathbb{R}^2$ applies when literal is ordinally-compared with column $\mathbf{c}$ (with op $=, >, \geqslant, <, \leqslant$). If op is $<$ or $\leqslant$ then $f_{\text{comp}}(m) = [0, f_{\text{scaled}}(m)]$. If op is $>$ or $\geqslant$, then $f_{\text{comp}}(m) = [f_{\text{scaled}}(m), 1]$. Finally, if op is $=$, then $f_{\text{comp}}(m) = [f_{\text{scaled}}(m), f_{\text{scaled}}(m)]$.

- ($f_4$) $f_{\text{ASCII}}(s) = [\texttt{ord(s[0])}\ \texttt{ord(s[1])},\ \texttt{ord(s[2])}] \in \mathbb{R}^3$. Applies to string literals, where $\texttt{ord(.)}$ is the ASCII code of character $\texttt{s[.]}$.

$(f_5)$ $f_{\text{date}}(d) = [\texttt{d.year}, \texttt{d.month}, \texttt{d.day}] \in \mathbb{R}^3$. Applies to date literals.

$(f_6)$ $f_{\text{tableSize}}(\texttt{table}) = \texttt{table.size} \in \mathbb{R}^1$. Applies for table nodes.

$(f_7)$ $f_{\text{columnRange}}(\mathbf{c}) = [\mathbf{c}.\texttt{minVal}, \mathbf{c}.\texttt{maxVal}] \in \mathbb{R}^2$. Applies for column nodes.

$(f_8)$ $f_{\text{ordinalOp}}(op) \in \{0,1\}^3$. Applies to ordinal operations $=, >, \geqslant, <, \leqslant$, respectively as $[010]$, $[001]$, $[011]$, $[100]$, $[110]$.

We leave the design of more intricate $f$'s as future work. The **learning features**

$$\mathcal{F} \subset \{(t, a, f) \mid (t, a) \in \mathcal{A}, \ f \in (\{0,1\}^* \to \mathbb{R}^*)\}, \tag{27}$$

allow us to customize how to extract numeric features from attribute $a$ node type $t \in \mathcal{T}$.

## G  EXPERIMENTS, ABLATION STUDIES, DISCUSSIONS

For ablation studies, we run experiments on CardBench workloads with increasing complexity, these datasets are downloaded from benchmark Chronis et al. (2024).

### G.1  HIERARCHICAL MODELS

We first examine the effectiveness and necessity of keeping multiple hierarchies in LITECARD. Table 5 compares the Q-Error metrics of different hierarchy configurations (using various combinations of $\mathcal{H}_1, \mathcal{H}_2, \mathcal{H}_3$) against POSTGRESQL on several CardBench datasets. The table shows that progressively incorporating more granular hierarchy levels ($\mathcal{H}_3, \mathcal{H}_2$, then $\mathcal{H}_1$) consistently improves estimation accuracy across datasets and percentiles. For instance, on 'cms' workload, the P90 Q-error improves from 112 (Postgres) to 110 ($\mathcal{H}_3, \texttt{P}$), then to 46.67 ($\mathcal{H}_2, \mathcal{H}_3, \texttt{P}$), and finally to 20.10 ($\mathcal{H}_1, \mathcal{H}_2, \texttt{P}$) or ($\mathcal{H}_1, \mathcal{H}_2, \mathcal{H}_3, \texttt{P}$). These results demonstrate the effectiveness of our hierarchical models in leveraging historical data to enhance the cardinality estimation capabilities of traditional optimizers. Moreover, Table 5 shows the need for multiple hierarchies. Comparing $(H_1, \texttt{P})$, $(\mathcal{H}_1, \mathcal{H}_2, \texttt{P})$, $(\mathcal{H}_1, \mathcal{H}_2, \mathcal{H}_3, \texttt{P})$, the latter two consistently outperform the first. This indicates that a simple hierarchy $(\mathcal{H}_1, \texttt{P})$ is insufficient, highlighting the importance of multi-level hierarchies.

### G.2  MODEL CHOICE

Figure 15 presents 50th percentile Q-errors comparing learned models (Linear Regression variants, Gradient Boosting, Gaussian Kernel) across hierarchy levels and datasets. Lower Q-errors are greener. The heatmap shows Gradient-Boosted Decision Trees (GBDT) achieve lowest median Q-errors, indicating superior accuracy. GBDT's E2E time is 49895s in Table 3, adding an overhead much smaller than savings due to better-optimized plans. Combined with efficient inference, GBDT was selected as the primary learner for LITECARD's overall evaluation (Table 3, Table 5).

### G.3  HISTORY SIZE

Figure 5 shows the impact of accumulated history size on LITECARD's estimation accuracy (P50 and P90 Q-Errors) on the IMDb workload. History size is less than or equal to x-axis value. The figure clearly shows that both P50 and P90 Q-Errors decrease significantly as the history size increases, especially in the initial stages. For instance, the P90 Q-Error drops sharply from over 200 towards 100 as history accumulates. The error curves then flatten, indicating that accuracy stabilizes once sufficient data is gathered for a template. This directly validates that LITECARD's learned models become more accurate as they are exposed to more examples through online learning.

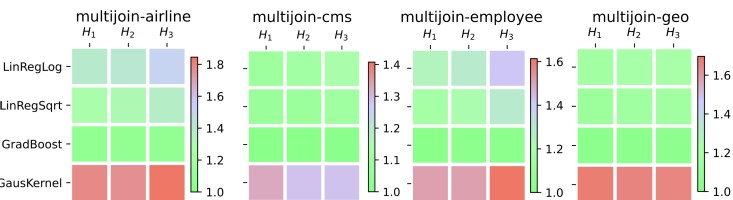

Figure 15: P50 Q-Error per database, comparing templatization strategies and learners.

## G.4 ESTIMATOR RELIANCE SHIFT WITH ACCUMULATED HISTORY

Figure 9 shows the proportion of subquery estimates from learned models vs. base POSTGRESQL as cumulative processed queries (history) increase on the 5k IMDb workload. The figure clearly demonstrates reliance shifting from POSTGRESQL (decreasing proportion) towards learned models (increasing proportion) as more history is gathered. This confirms LITECARD's online learning effectively leverages history to replace base estimates, underpinning iterative performance gains (Figure 8).

# H RUNTIME ANALYSIS

**Minimal Training Overhead Enables Online Learning.** Table 3 and Figure 3 presents the total training overheads for all learned techniques. Offline, batch-trained methods like MSCN, DEEPDB, FACTORJOIN, and fine-tuned PRICE incur substantial overheads, ranging from 1,466 seconds (MSCN) to 14,828 seconds (PRICE fine-tuned). Note these exclude data collection costs for query-driven methods $\approx$ 34 hours for MSCN). Such high costs impede frequent updates. In contrast, LITECARD, an online learner, starts with zero initial overhead and incurs a total training overhead of only 37.29s for the 5k workload via lightweight incremental updates ($\approx$ 0.001s each). These updates can be performed asynchronously.

This minimal overhead enables practical online learning and continuous adaptation, fundamentally distinguishing LITECARD from expensive batch retraining paradigms.

## H.1 DETAILED ANALYSIS

**Detailed Runtime Comparison.** Figure 7 shows the relative End-to-End time improvement over POSTGRESQL (0% line) for queries grouped by their original PG runtime. For very short queries ([0-0.008s], [0.008-0.66s]), most learned methods show degradation, as optimization time dominates. PRICE exhibits the largest degradation, while LITECARD stays close to POSTGRESQL and even shows a slight initial improvement. For longer queries (especially >200s), where execution time is substantial, learned methods like DEEPDB, FACTORJOIN, and LITECARD achieve significant improvements, as the benefit of better estimates outweighs optimization overhead. This demonstrates that low optimization overhead is crucial for performance on short queries, while estimation accuracy drives improvements on long ones. Figure 7 confirms LITECARD provides robust performance across query runtimes, avoiding degradation on short queries due to its low optimization cost, while delivering substantial gains on long queries.

**Relative Estimation Error Distribution.** Figure 6 shows the distribution of relative estimation errors (estimated/true) for all 46,928 subqueries on the 5000-query IMDb workload. Perfect estimates are at 1. The figure reveals POSTGRESQL and PRICE estimates are heavily skewed below 1, indicating significant underestimation bias. In contrast, LITECARD, DEEPDB, FACTORJOIN, and MSCN distributions are centered around 1, showing reduced bias. LITECARD and DEEPDB exhibit the tightest distributions around 1, signifying lower error variance. Such reduced bias and variance are crucial for effective query optimization. Figure 6 demonstrates LITECARD significantly improves estimation accuracy and reduces the underestimation bias compared to PostgreSQL.

**Iterative Improvement through Online Learning.** Figure 8 shows LITECARD's End-to-End time over 5 iterations on the first 1000 IMDb queries, compared to static baselines. LITECARD demonstrates a clear performance improvement trend, decreasing from $\approx 11,200$ seconds at Iteration 1 to $\approx 9,500$ seconds by Iteration 5. It starts faster than POSTGRESQL and MSCN, matches FACTORJOIN and PRICE early, and approaches DEEPDB and ORACLE performance over time. This improvement stems from effective online learning, where LITECARD refines its models with each processed query. Figure 8 demonstrates that LITECARD's online learning delivers iterative End-to-End performance improvements, allowing it to adapt and become increasingly competitive with static learned estimators.

Table 5: Q-Error Comparison on CardBench Workloads.

| Model | cms | | | stackoverflow | | |
|---|---|---|---|---|---|---|
| | $Q_{\text{err}}^{50}$ | $Q_{\text{err}}^{90}$ | $Q_{\text{err}}^{95}$ | $Q_{\text{err}}^{50}$ | $Q_{\text{err}}^{90}$ | $Q_{\text{err}}^{95}$ |
| Postgres | 3.33 | 112 | $2.3e^3$ | 4.85 | 360 | $3.1e^3$ |
| $(H_3, \text{P})$ | 3.21 | 110 | $2.2e^3$ | 4.30 | 367 | $3.8e^3$ |
| $(H_2, H_3, \text{P})$ | 1.15 | 46.67 | 159 | 1.16 | 44.33 | 464 |
| $(H_1, \text{P})$ | 1.07 | 22.22 | 97.00 | 1.12 | 21.03 | 200 |
| $(H_1, H_2, \text{P})$ | **1.06** | **20.10** | **94.48** | **1.11** | **18.01** | **182** |
| $(H_1, H_2, H_3, \text{P})$ | **1.06** | **20.10** | **94.48** | **1.11** | **18.01** | **182** |

| Model | accidents | | | airline | | |
|---|---|---|---|---|---|---|
| | $Q_{\text{err}}^{50}$ | $Q_{\text{err}}^{90}$ | $Q_{\text{err}}^{95}$ | $Q_{\text{err}}^{50}$ | $Q_{\text{err}}^{90}$ | $Q_{\text{err}}^{95}$ |
| Postgres | 1.65 | 10.31 | 18.29 | 1.63 | 97.30 | 216 |
| $(H_3, \text{P})$ | 1.34 | 8.93 | 20.60 | 1.59 | 97.00 | 216 |
| $(H_2, H_3, \text{P})$ | 1.15 | **4.81** | **15.42** | 1.20 | 13.88 | 91.00 |
| $(H_1, \text{P})$ | 1.15 | 4.95 | 17.25 | 1.13 | 4.50 | 29.20 |
| $(H_1, H_2, \text{P})$ | **1.15** | 5.02 | 17.70 | **1.13** | **4.29** | **25.00** |
| $(H_1, H_2, H_3, \text{P})$ | **1.15** | 5.02 | 17.70 | **1.13** | **4.29** | **25.00** |

| Model | employee | | | geo | | |
|---|---|---|---|---|---|---|
| | $Q_{\text{err}}^{50}$ | $Q_{\text{err}}^{90}$ | $Q_{\text{err}}^{95}$ | $Q_{\text{err}}^{50}$ | $Q_{\text{err}}^{90}$ | $Q_{\text{err}}^{95}$ |
| Postgres | 1.54 | 3.38 | 4.83 | 224 | $2.1e^5$ | $1.2e^6$ |
| $(H_3, \text{P})$ | 1.35 | 3.14 | 4.42 | 218 | $2.1e^5$ | $1.2e^6$ |
| $(H_2, H_3, \text{P})$ | 1.05 | 2.11 | **2.98** | 1.10 | $5.8e^3$ | $7.3e^4$ |
| $(H_1, \text{P})$ | 1.03 | 2.09 | 3.07 | 1.09 | 192 | $1.1e^4$ |
| $(H_1, H_2, \text{P})$ | **1.03** | **2.03** | 3.01 | **1.08** | **66.38** | **7.0e$^3$** |
| $(H_1, H_2, H_3, \text{P})$ | **1.03** | **2.03** | 3.01 | **1.08** | **66.38** | **7.0e$^3$** |

