# OpenReview forum: "Is it Bigger than a Breadbox: Efficient Cardinality Estimation for Real World Workloads"
_ICLR.cc/2026/Conference — Submitted to ICLR 2026_

### Official Review · Reviewer_FAp8 · 2025-10-27

**Soundness:** 2
**Presentation:** 2
**Contribution:** 2
**Rating:** 2
**Confidence:** 3

**Summary:**

The paper introduces a cardinality estimation for SQL queries based on kernel regression and handcrafted features.
The method has quite small overhead and competes with SoTA learning-based approaches on error metrics.
Further, amending PostgreSQL with the method achieves notable accuracy and runtime improvements over traditional methods and drastically reduces operational costs compared to other learned cardinality estimators, thereby offering the most practical and efficient solution on the Pareto frontier.

**Strengths:**

1. The language of the paper is fine, main ideas are easy to follow.
2. Empirical results show that the proposed method has quite fast inference (not sure about training?), when compared to alternatives.

**Weaknesses:**

1. The performance is lower than some baselines (Table 3).
2. The equation (6) is not clear. In ML, one should use train/test split of a dataset to avoid overfitting.
3. Figure 5 caption: missing reference
4. It seems that your method is called "LITECARD". But it was never introduced formally.
5. The main idea of the proposed method is a specific vectorization of execution DAGs. The features (Appendix D) are very domain specific and lack generality. I propose to resubmit the manuscript to VLDB or KDD.
6. A comparison with standard GCN, GAT is not provided.
7. I can't find proofs that the proposed method  (i) can run from cold-start, requiring no upfront training; (ii) can adapt to changes in workloads or data shifts

**Questions:**

1. Can you compare your method with:
* P. Negi, R. Marcus, H. Mao, N. Tatbul, T. Kraska, and M. Alizadeh, ‘‘Cost-guided cardinality estimation: Focus where it matters,’’ in Proc. IEEE 36th Int. Conf. Data Eng. Workshops (ICDEW), Apr. 2020, pp. 154–157.
* J. Sun and G. Li, ‘‘An end-to-end learning-based cost estimator,’’ 2019, arXiv:1906.02560.
* J. Sun, J. Zhang, Z. Sun, G. Li, and N. Tang, ‘‘Learned cardinality estimation: A design space exploration and a comparative evaluation,’’ Proc. VLDB Endowment, vol. 15, no. 1, pp. 85–97, 2021

2. How your paper is related to:
Woltmann, L., Hartmann, C., Thiele, M., Habich, D., & Lehner, W. (2019, July). Cardinality estimation with local deep learning models. In Proceedings of the second international workshop on exploiting artificial intelligence techniques for data management (pp. 1-8) ?

3. It is not clear how the model is trained. Is it trained online or query dataset is divided into train/test splits?

---

> ### Author Response · Authors · 2025-11-20
>
> Thank you for your detailed review. Your comments led us to add new experiments (including train/test splits and GNN baselines) and to clarify several aspects of the method. We address each point below.
>
> ## Strengths
>
> * We would like to clarify that the reported overhead numbers include both training and inference time; see “System, Scale, and Efficiency” in the main response for a detailed breakdown.
>
> ## Weaknesses
> * [W1] We agree that our method does not always achieve the lowest Q-error among all deep baselines. Our goal, however, is to be on the practical accuracy–overhead Pareto frontier, not to dominate purely on accuracy. Other deep models do offer better performance sometimes, but they incur substantial optimization and runtime overhead, which makes deployment challenging in practice. For example, figure 4 shows the query optimization time for a 10-table join query with a learned CE can take on the order of ~1s, whereas the default PostgreSQL optimizer takes ~30ms. A ∼30× increase in optimization time is difficult to justify in real-world systems. In addition, existing methods (query-driven, data-driven, and zero-shot) all require either training data collection or data-dependent preprocessing when encountering a new dataset, which can take minutes to hours. In contrast, our lightweight local models can start from zero data and learn fully online. We will clarify this in the revised paper. Please also see the main response "Contribution and Novelty" Section.
> * [W2] Equation (6) is not clear: It is the multiplication of two terms -- a binary term (evaluates to 1 if graphs are isomorphic and to 0 otherwise) with a positive term quantifying similarity of 2 feature vectors (Gaussian kernel defined in Eq.4). Both the indicator notation and Gaussian kernel are standard in ML; we will add a short explanation to make this clearer.
>
>   Regarding train/test splits: you are right that standard ML practice uses fixed splits to avoid overfitting. Our primary setting is online learning, where examples arrive in sequence and the learner may use all past data when predicting the next query. This is only viable if training is extremely fast, which is exactly the case here (our local models can be updated in milliseconds).
>
>   That said, we agree that a “standard” batched evaluation is informative. In response to your comment, we have added experiments with fixed train/test splits on the benchmark datasets, and a GNN baseline (GCN-style) implemented on the same data. These results are summarized in the main response and will be incorporated into the revised paper.
> * [W3] Fig5: We have fixed it, and we will upload the updated manuscript.
> * [W4] We now mention LiteCARD since the abstract and intro, and we will upload the updated manuscript.
> * [W5] Our main methodological idea is a canonical ordering of nodes in a DAG: the feature vector would be exactly the same if node order is shuffled -- this is the bulk of the method, continuing to hierarchically partition queries according to increasingly granular features (Sec 3.5). This canonicalization + graph-local kernel is the core of the method. The concrete feature extractors we plug into this framework are summarized in the appendix and tailored to relational DBMSs.
>
>   More broadly, we present aspects of the paper that align with [ICLR's Call-for-Papers (cfp)](https://iclr.cc/Conferences/2026/CallForPapers), which explicitly lists topics:
>     * metric learning, kernel learning
>     * learning on graphs and other geometries & topologies
>
>     and also welcomes contributions related to software implementation:
>
>     * infrastructure, software libraries, hardware, etc.
>
>   If, after discussion among the reviewers and area chair, the committee feels that this work is better suited for a different venue, we will of course respect that decision. However, we believe that the combination of graph-local kernel learning, online adaptation, and empirical study on real DBMS workloads aligns well with the scope articulated in the ICLR CFP.
> * [W6] Thank you for raising this! We have now implemented a GNN that is a trivial extension of GCN onto heterogeneous graphs. We pasted the results up top.

---

> ### Author Response · Authors · 2025-11-20
>
> * [W7] Cold start and adaptation to workload/data shifts.
>
>   The proposed method satisfies (i) & (ii) by design. LiteCard is an online estimator:
>     * At startup, the history $\(\mathcal{D}\)$ in Eq. (5) is empty and the hash table of patterns has no entries. For each new query, PostgreSQL produces a plan with subqueries; we compute each subquery DAG's pattern hashes $\(h^H(G)\)$, and try to look up a local model. If no model exists for that hash, we fall back to PostgreSQL’s native estimator, so the system is immediately usable from the very first query with zero upfront training.
>     * As queries execute, the executor provides the true cardinalities. For each pattern, we update its tiny local model in **milliseconds per update**. Thus, repeated patterns quickly gain accurate learned estimates, while unseen patterns retain the safety of PostgreSQL.
>     * Of course, the inference seems slow (having to loop through all history for every inference). However, the specific instantiation of the kernel-across-graphs of Eq.6 being indicator-times-value allows us to just lookup the graphs for which the indicator evaluates to 1 (implemented as a hash-table lookup). This makes online learning efficient.
>    There is no separate offline training phase and no requirement for pre-existing logs: learning begins with the first query and improves as the workload runs.
>
>   For adaptation to workload or data shifts, the same mechanism applies, with three common scenarios:
>
>    - New query templates (workload shift): New subquery structures simply hash to new keys. At first, these patterns fall back to PostgreSQL (safe behavior). After a few occurrences, their local models are trained online and begin providing improved estimates. No special “retraining phase” is needed—adaptation is just continued online updates.
>
>    - Data distribution shift for existing templates: For a fixed pattern hash $\(h(G)\)$, we keep adding new $\((x(G), y(G))\)$ pairs to its history and update the local regressor ${g_\theta(G)}$ (Eq. (5)). Because updates are cheap, we can update on every query; as the distribution shifts, new labels dominate and the model naturally tracks the new regime.
>
>    - Schema changes / major regime changes: When the schema or workload changes significantly, we can clear (or prune) the hash table and let it repopulate from new queries. Since each local model is tiny and training is millisecond-scale, “retraining from scratch” on the new regime is inexpensive.
>
>   In all cases, no extra offline retraining, data collection, or statistics rebuilding is needed for adaptation. We will clarify this in the revised paper.
>
> # Questions
> * Q1&2: We have discussed each paper in the main response. Please refer to the "Additional Related Work" section.
> * Q3: The model is trained on-the-fly, at every query: Eq.5 contains $\arg\min$ i.e. model training. However, it also makes sense to provide an apples-to-apples comparison. Therefore, we ran experiments controlling the train/test sets and we pasted their results in the main responses. Please refer to the "Additional Experiments and Observations" section.

---

> > ### Comment · Reviewer_FAp8 · 2025-11-26
> > **Response**
> >
> > Thank you for a detailed answer. Many of my questions are answered.
> > Can you please explain the idea of "canonical ordering"? It is very hard to understand from the paper.
> > Why is the ordering canonical? Also, the hashing algorithm is hard to understand too.
> > I recommend to use a running example with a simple query and explain hashing/canonical ordering with the example.
> > Also, I suppose that for cardinality estimation real-valued parameters are important.
> > A cardinality of a query with a condition X>10 will be close to X>11.
> > But with hashing the notion of closeness is lost.
> > How do you solve this issue?

---

> ### Author Response · Authors · 2025-11-28
>
> ## Canonical node ordering and step-by-step example
>
> Canonical ordering means that the feature vector, e.g., for the query you mention (X>10) it could be: ['>', 10], regardless of order[*] of nodes e.g., (X, >, 10)  vs (>, X, 10). You may look at Figure 10, to see how all (orange, blue, red) queries always extract features in a canonical  node order.
>
> For determining node ordering, only features in \mathcal{H} are considered. Once ordering is established, features named in \mathcal{F} will be extracted from graph nodes visited in that order. Suppose hyperparameters \mathcal{H} = {(column, name)} and let[**] \mathcal{F} = {(literal, value), (op, code)}   -- see Table 1 for more examples, then, the steps are:
>
> 1. The graph (with features) is:
>     ```
>     (column {name: X})    -->     (op {code: >})     <--     (literal {value: 10})
>     ```
> 2. Node-wise hashes will be computed, initializing the $\mathbf{H}$ array of strings (per Eq.8) as:
>     ```
>     H = [hash("column {name: X}"),  hash("op {}"), hash("literal {}")]
>     ```
>     At this point, each node is hashing only its features intersecting with features named in \mathcal{H}.
> 3. Then, each node updates itself given all its predecessors and successors, which can be done in 2 topological order passes -- luckily, our input graphs are DAGs by construction,  guaranteeing a valid topological order. (Eq.9 and 10).
>
>     Suppose, after applying Eq.9 and 10,  that
>     ```
>     H=['1A001...', 'F4D4..', '003A..']
>     ```
> 4. Eq.11 would set `π = [2, 0, 1]`
> 5. Eq.12 follows ordering of `π`  extract feature vector [10, '>']  (per \mathcal{F})
>
> As we type these steps, we realize that it is not immediately obvious from reading he paper and we will be adding a figure (in main paper or appendix) to explain this.
>
> [*] Edges are labeled as "rhs" vs "lhs", for asymmetric binary ops. We plan to open-source the SQL->Graph parser.
> [**] In reality, there can be additional features coming from the column statistics of X, or features from its source table -- note: we dont use any Postgres features, leaving as future work.
>
>
>
> ## Hashing-away literals
>
> Your intuition is spot-on. The literal value **is** the most important part of the query that would be used for cardinality estimation. This is why we always extract this feature for all settings (in Table 1, right column \mathcal{F} always includes {(literal, value)}). Crucially, the entry {(literal, value)} is **not** used for hashing (in Table 1, it is never on the left column \mathcal{H}).
>
> This implies that "X>10" and "X>11" **are structurally similar**, i.e., isomorphic once features are intersected with \mathcal{H}, with indicator function of Eq.6 evaluating to 1.
> Their final similarity would be equal  to the euclidean distance of vector [10] and vector [11], *i.e.* = 1, raised to negative exponential (per Eq.4).

---

### Official Review · Reviewer_zkyh · 2025-10-28

**Soundness:** 3
**Presentation:** 2
**Contribution:** 2
**Rating:** 4
**Confidence:** 4

**Summary:**

This paper presents LITECARD, an online, pattern-based learned cardinality estimator for relational databases. The key idea is to exploit the repetitive nature of sub-query patterns in real-world workloads by grouping isomorphic subquery graphs and training lightweight regressors (e.g., locally weighted linear regression or decision forests) per pattern.

**Strengths:**

S1. Practical motivation: The paper addresses a genuine gap in learned cardinality estimation research, bridging the accuracy of learned models with the low overhead needed for production DBMS integration.

S2. Novel technical framing: The use of graph-local learning combined with hash-based partitioning of isomorphic subqueries is original and conceptually neat. The hierarchical (H1–H3) partitioning and incremental online updates are well-justified.

S3. System integration: Implementing the method in PostgreSQL and demonstrating measurable runtime improvement (e.g., 7.5 min, around 30% reduction on IMDb) is good.

**Weaknesses:**

W1. Unclear pattern detection pipeline.
It is unclear how the query patterns are discovered or clustered online. The paper seems to assume that the graph of subqueries is already extracted from query plans, but does not explain whether clustering or pattern identification contributes to the overhead. Clarifying the computational cost and scalability of this step is essential.

W2. Dependence on historical query plans.
The method appears to rely on a corpus of historical queries and their plans to build pattern-specific models. This raises several questions:

- Are the query plans generated by PostgreSQL itself?

- Is it fair to compare against methods that do not assume access to such historical execution traces?

- How is the cold-start phase (when no history exists) handled?

W3. Generalization and workload shift.
Since models are trained per pattern derived from past workloads, the approach seems inherently query-driven. How well does it transfer to new workloads or schema variations? If a new workload introduces unseen patterns, does the system re-hash and retrain? Are such adaptation costs included in the reported “37 s overhead”? Clarifying this is crucial for evaluating the claim of “negligible overhead.”

W4. Scalability with large workloads.
The method’s efficiency is evaluated on IMDb (about 5 k queries). How does performance scale when the number of query patterns grows (e.g., 50 k queries or more)? The memory cost of maintaining many regressors and hash tables may become significant, while data-driven baselines remain unaffected by query count.

W5. Experimental scope and baseline selection.
The evaluation focuses primarily on IMDb. It would strengthen the paper to include other complex benchmarks (e.g., STATS, StackOverflow, or TPC-DS) that feature multi-table joins and diverse predicates.
Also, the baselines are limited to DeepDB and FactorJoin as “data-driven” methods, but several  approaches (e.g., NeuroCard 2021, FLAT 2021, CardBench baselines) could provide a fairer SoTA comparison. The authors should justify why these were omitted.

Minor issues.

Figure 5 caption contains a typo: “Eq. ??” should reference the correct equation number.

The paper sometimes intermixes the terms LITECARD and ours; consistent terminology would improve clarity.

**Questions:**

Q1. How are query patterns identified? Is there an explicit clustering step, and is its cost part of the 37 s overhead?

Q2. How are query plans obtained for historical queries? If the system requires pre-existing plans, does that imply an extra source of information unavailable to other baselines?

Q3. How does LITECARD handle new query workloads or schema changes where the subquery graphs differ from historical ones?

Q4. How does performance and overhead scale as the number of unique query patterns increases (e.g., beyond 50 k)?

Q5. Why is IMDb the only dataset tested? Would the method generalize to datasets with more complex join paths or string predicates (e.g., STATS, TPC-DS)?

Q6. Why were only DeepDB and FactorJoin chosen as data-driven baselines, and are they considered current SoTA for this comparison?

---

> ### Author Response · Authors · 2025-11-20
>
> Thank you for your detailed and insightful comments. They greatly helped clarify our presentation and improve the manuscript. Below, we address each point.
>
> ## Weaknesses
>
> * [W1 & Q1] Pattern detection pipeline and overhead.
>
>     * We do not change the Query Planner of Postgres (which is similar to many other DB engines). We only integrate with it. As summarized in Section 2.1 (Background), the planner already decomposes a query into many subqueries, on which we perform learning.
>     * The patterns are extracted as described in section 3.2: a pattern is the hash $h^H$ of Eq.11 computed from the subquery DAG.
>     * Hashing time is negligible, compared to training and inference of the light-weight models: yes, it is part of the 37 seconds, as broken down in the "System, Scale, and Efficiency" Section of Main Response.
>     * We will make this pipeline explicit in the revised manuscript and include an overhead breakdown (hashing, feature extraction, model inference, model update).
>
> * [W2 & Q2]: Dependence on historical query plans.
>     Our method does not rely on any pre-existing history. Learning and inference are performed online:
>
>     * Query plans are generated by PostgreSQL during normal execution; we reuse them as they appear online.
>     * Learning begins from the very first query.
>     * If a pattern has never been seen before, LiteCARD automatically falls back to PostgreSQL’s estimator — this is the cold-start behavior.
>     This is different from query-driven methods that require historical traces or offline training.
>     Existing deep learning baselines **cannot** use this information **on-the-fly**. For an apples-to-apples comparison, we also ran experiments with fixed training and test sets (ignoring the "online advantage" of our method); these are reported in the additional experiments section in the main response. Nonetheless, the training overhead time to use such information is non-negligible: training a plain GNN takes 5–10 minutes on a small portion of the data, and SoTA models such as DeepDB take hours, whereas each LITECARD update takes only milliseconds.
>
> * [W3 & Q3] Generalization to workload shifts or schema changes.
>   Thank you for highlighting this point. LITECARD supports workload evolution by design:
>
>   * If a workload introduces new patterns, the system simply creates new hash buckets and trains new regressors on-the-fly.
>   * If schema changes cause major structural shifts (eg. schema change), the system can reset the hash table (or keep the last K samples per pattern).
>   * Because each local model is extremely light (~ms to train/update), adaptation cost is essentially negligible.
>
>   The reported 37s overhead includes all hashing, feature extraction, and model updates for the full 5k-workload; shifts would incur similarly small per-query costs. We will add these clarifications.
>
> * [W4 & Q4] Scalability with large workloads.
>
>   * We keep one small model per pattern, so memory scales linearly with the number of patterns. In most real workloads, queries are generated by applications, and query templates repeat heavily. A large Redshift fleet study van Renen et al., VLDB’25[1] shows that >95% of queries repeat in same template within a month (see figure 5c), so a 50k-query workload typically corresponds to ~2k distinct templates.
>   * Lookup and model-selection overhead is logarithmic in the number of patterns (hash map + red-black tree behavior).
>   * If a workload truly has an unbounded stream of new human-written patterns, one can employ standard model-management strategies (e.g., LRU eviction, tiered caches). We will mention these “how to scale further” strategies in the future work section.
>
>   We also refer you to the “System, Scale, and Efficiency” section of the main response, where we break down the 37s overhead and show that the bulk of the time (~34s) comes from synchronous model updates, while hashing and inference are negligible. In a deployment, updates can be moved to a background thread without changing the learning algorithm.
>
> * [W5 & Q5] Following your suggestion, we have conducted new experiments on datasets in CardBench (which include workloads with multi-table joins and diverse predicates). These results are summarized in the “Additional Experiments” section of the main response and will be incorporated into the revised manuscript.
>
> * [W5 & Q6]. We selected a couple of representative methods that were published recently. We further responded in the main response section, "Additional Related Work". Please see it there.
>
> [1] van Renen et al. "Why TPC Is Not Enough: An Analysis of the Amazon Redshift Fleet", VLDB 2025, https://www.vldb.org/pvldb/vol17/p3694-saxena.pdf

---

> ### Comment · Reviewer_zkyh · 2025-11-24
>
> The reviewer thanks the response. It do save part of my concerns. Overall, using pattern clustering and training a proxy model for each cluster seems intuitive. The high-level idea is interesting. However, the reviewer is still not clear that how a pattern is generated, even with equation 11 given. The reviewer strongly suggests to add a concrete example explaining this since it is part of the main contributions.

---

> ### Author Response · Authors · 2025-11-28
>
> Since another reviewer asked a question pertaining this, we officially admit that a visualization diagram is needed to explain these steps. Taking the example of reviewer **FAp8**:
>
> Suppose a query "X > 10", then, regardless of order[*] of nodes e.g., (X, >, 10)  vs (>, X, 10), we want to establish a canonical node order. It is useful for (1) feature extraction (to be in a canonical order) and (2) for computing a hash value of the subgraph (Eq10).  Figure 10 highlights that features for all queries (orange, blue, red) are consistently ordered.
>
> For determining node ordering, only features in \mathcal{H} are considered. Once ordering is established, features named in \mathcal{F} will be extracted from graph nodes visited in that order. Suppose hyperparameters \mathcal{H} = {(column, name)} and let[**] \mathcal{F} = {(literal, value), (op, code)}   -- see Table 1 for more examples, then, the steps are:
>
> 1. The graph (with features) is:
>     ```
>     (column {name: X})    -->     (op {code: >})     <--     (literal {value: 10})
>     ```
> 2. Node-wise hashes will be computed, initializing the $\mathbf{H}$ array of strings (per Eq.8) as:
>     ```
>     H = [hash("column {name: X}"),  hash("op {}"), hash("literal {}")]
>     ```
>     At this point, each node is hashing only its features intersecting with features named in \mathcal{H}.
> 3. Then, each node updates itself given all its predecessors and successors, which can be done in 2 topological order passes -- luckily, our input graphs are DAGs by construction,  guaranteeing a valid topological order. (Eq.9 and 10).
>
>     Suppose, after applying Eq.9 and 10,  that
>     ```
>     H=['1A001...', 'F4D4..', '003A..']
>     ```
> 4. Eq.11 would set `π = [2, 0, 1]`
> 5. Eq.12 follows ordering of `π`  extract feature vector [10, '>']  (per \mathcal{F})
> 6. The final hash would be:
>     ```
>     h = hash(H[π[0]] + H[π[1]] + H[π[2]])
>     ```
>     which captures all node features, as well as their connections.
>
> [*] Edges are labeled as "rhs" vs "lhs", for asymmetric binary ops. We plan to open-source the SQL->Graph parser.
>
> [**] In reality, there can be additional features coming from the column statistics of X, or features from its source table -- note: we dont use any Postgres features, leaving as future work.

---

### Official Review · Reviewer_NayK · 2025-10-29

**Soundness:** 3
**Presentation:** 3
**Contribution:** 3
**Rating:** 6
**Confidence:** 3

**Summary:**

This manuscript delineates an innovative approach to cardinality estimation called LITECARD, which addresses the trade-off between the inaccurate, fast heuristics of traditional database systems (like PostgreSQL's per-column histograms) and the accurate, slow deep learning models that are often too complex to deploy in practice. The LITECARD works by decomposing the query space and applying graph-local learning, which exhibits an amalgamation of pattern recognition, custom kernel, local models, and hierarchical fallback, a combination that endows the model with superior performance in comparison to baseline methods.

**Strengths:**

S1. The paper's significance is underscored by its contribution of a novel cardinality estimator that 1) can run from a cold-start (no upfront training), 2) can adapt to changes in workloads or data shifts, and 3) has negligible update and inference time.

S2. The estimator's novelty is encapsulated in its graph-local learning - an amalgamation of pattern recognition, custom kernel, local models, and hierarchical fallback.

S3. The evaluation is comprehensive, with comparisons to baseline methods providing a compelling demonstration of the superior performance of the proposed method.

**Weaknesses:**

W1. While the median error (P50) is highly competitive (1.70), the tail errors (worst-case errors) are significantly worse than the most accurate learned estimators, DeepDB and FactorJoin.

W2. The core efficiency and low overhead rely on the assumption that highly repetitive subquery patterns exist in the workload. For a workload with extremely low query template reuse or high churn in unique query patterns, the model would frequently fall back to the base PostgreSQL estimator or have to train local models on very sparse data, undermining its advantage.

W3. The local learning models are simple and established. While simplicity contributes to the low overhead, relying on models that only consider local feature proximity might limit their ability to capture complex non-linear feature interactions within a query pattern, potentially contributing to the high tail errors. The novelty rests more on the system-level integration and workload decomposition strategy rather than the machine learning models themselves.

**Questions:**

See "weaknesses" above.

---

> ### Author Response · Authors · 2025-11-20
>
> Thank you very much for the thoughtful review and for highlighting the key strengths of our work . Below we address all raised weaknesses in detail.
>
> ## Responses to Weaknesses / Questions
>
>   * [W1] Yes, in terms of accuracy, our method is competitive with—but not always better than—deep neural network models. The higher tail errors arise primarily when a pattern has very limited historical data. As shown in Figure 5, these tail errors (e.g., P90) decrease steadily as the history size increases.
>
>      Importantly, our objective is to sit on the practical accuracy–overhead Pareto frontier:
>
>      * deep models achieve stronger P90 at the cost of hours of preprocessing/training,
>      * while LiteCard achieves substantial end-to-end speedups (+27%) with only 37s total overhead.
>
>     Nonetheless, it is **possible** to do something in-between, e.g. consider the two options:
>
>       1. Cluster queries based on graph isomorphism (like indicator function of Eq.6 of our paper), but then set $g_\theta$ (Eq.5) to something more complex than decision forest or locally-weighted linear regression, e.g., 2- or 3-layer MLP.
>       2. Like Section **3.5 Hierarchical Data Structure**, but replace the last term of the section "traditional cost estimator" (referring to Postgres) with a more advanced model, e.g., DeepDB or FactorJoin, only in cases where the DB system engineers are willing to update the models as database contents or workload patterns shift.
>
>     * We view these extensions as promising future work and will add a short discussion in the revised manuscript.
>   * [W2] You mentioned that **"core ... rely on the assumption that highly repetitive subquery patterns exist"**. Thank you for pointing this out! Our method is indeed designed to exploit real-world template repetition, and this assumption is strongly supported empirically:
>
>     According to a recent large-scale Amazon Redshift fleet study [1] -- see their Figure 5c:
>       * for more than 50% of the database clusters, more than 80% of queries have already appeared *in template* in the same day.
>       * more than 95% of queries have already appeared *in template* in the same month.
>
>     We now cite these results explicitly in the manuscript.
>   * [W3] We appreciate the opportunity to clarify this. While local models are very established [since decades], graph-local methods -- where the similarity function operates on two graphs -- are in fact much more rare. While the system-level integration is novel indeed (of appendix), we also feel that the main paper is novel on its own (without appendix), including graph-local learning, canonical graph hashing and feature-order alignment guarantees, background/context, theorem ideas of why the algorithms work, and experiments demonstrating usefulness, including the rebuttal-added experiments. While our PostgreSQL integration is indeed a system contribution, the core algorithm itself is new, and we have strengthened this explanation in the revised manuscript's intro section.
> ### Reference
> [1] van Renen et al. "Why TPC Is Not Enough: An Analysis of the Amazon Redshift Fleet", VLDB 2025, https://www.vldb.org/pvldb/vol17/p3694-saxena.pdf

---

### Official Review · Reviewer_AwTu · 2025-11-01

**Soundness:** 3
**Presentation:** 3
**Contribution:** 3
**Rating:** 4
**Confidence:** 3

**Summary:**

The paper proposes a lightweight online learning approach for query cardinality estimation in relational databases. Instead of using a single large model, it maintains multiple small regressors, each specialized for a subquery pattern represented as a graph and retrieved via hashing. The method continuously updates online with minimal overhead and integrates into PostgreSQL, achieving accuracy comparable to state-of-the-art learned estimators while reducing training cost and runtime overhead.

**Strengths:**

1. The problem studied in this paper is important.
2. The presentation is good.

**Weaknesses:**

1. Unclear novelty. The paper does not clearly explain how it differs from existing learned optimizers and cardinality estimation methods.
2. Insufficient scalability analysis. The paper lacks detailed discussion and evaluation of how the method scales with larger datasets and workloads.

**Questions:**

I have two main concerns about the paper: novelty and scalability.

1. Novelty
The paper discusses several existing learned optimizers and cardinality estimation approaches. However, the fundamental novelty of this work remains unclear. Please clarify what distinguishes the proposed method from prior works — for example, what new information or mechanisms are introduced in the learning process, and how these lead to fundamentally different behavior or advantages compared with existing models.

2. Scalability and Training Overhead
Section 4.1 reports a total overhead of about 37 seconds for 5k queries on the IMDb dataset, but it is unclear how this overhead scales with larger workloads. Specifically:
	•	How does the system perform as the number of stored subquery patterns or regressors increases (e.g., 100k queries or multi-terabyte datasets)?
	•	Does the memory footprint or lookup latency grow linearly with the number of hashed entries?
	•	Since each pattern maintains a separate model, could model management or hash collisions become a bottleneck for very large workloads?

---

> ### Author Response · Authors · 2025-11-20
>
> Thank you very much for your thoughtful comments. We address both novelty and scalability concerns below, and we will incorporate all clarifications into the revised manuscript.
>
> ## Response to Novelty
> In summary, our method learns on-the-fly (compared to other methods that employ deep networks that need to be trained and/or fine tuned as workload pattern shifts). The reason why this works very well, in practice, is due to observations in a recent [Redshift paper](https://www.vldb.org/pvldb/vol17/p3694-saxena.pdf) [1] -- see their Figure 5c: in their actual system, for more than 50% of the database clusters, more than 80% of queries have already appeared *in template* in the same day, and more than 95% have already appeared *in template* in the same month.
>
> Please also refer to **Contribution and Novelty** in the main response.
>
> We now added a clearer paragraph in the introduction for better presentation: "Our contributions are ..."
>
>
>
> ## Response to System scale:
> We presented a breakdown of the 37 seconds (please see **"System, Scale, and Efficiency"** of Main Response).
> Importantly, this 37s includes all components: hashing, lookup, model inference, and model updates.
>
> The vast majority of the time (~34 s) comes from model updates, because in our experimental setup we update the decision forest synchronously on the main thread for every query to simplify measurement. Hashing and inference together take only a negligible fraction of the total time.
> In a live deployment, these model updates can naturally be moved to a background or asynchronous thread, which would reduce the impact on query latency without modifying the learning algorithm.
>
> Onto your other questions:
>
> * Our method scales ~linearly with the number of patterns. We need to keep one model (e.g., decision tree) per pattern. This is fine in most use-cases, where the queries are usually generated by a computer program (hence following a closed-set of patterns). In cases where there are a stream of new patterns (e.g., by human data analysts), then the one must worry about model management, e.g., adopt some LRU (least-recently-used) eviction strategies.
>
> * Practical number of patterns: as shown in Redshift [1], ~95% of queries appear in the same template within the same month, meaning that for a 100k queries workload it typically corresponds to ~5k unique  patterns due to the template repetition. Thus the effective number of models is modest.
>
> * As number of hashed entries $(n)$ grow, the memory footprint should grow linearly as $O(n)$ for decision trees and $O(n * m)$ for locally-weighted linear regression (or one-shot classifiers, i.e., RBF), where $m$ is the average dimensionality of the feature vector extracted from (sub)queries. Since we use the default hashmap implementation (of C++ for Postgres, and of Python for Q-Error analysis), the worst-case time complexity is $O(\log n)$ -- [ref: Red Black Trees](https://en.wikipedia.org/wiki/Red%E2%80%93black_tree)
>
> We will add the breakdown to the main paper and mention the "how-to-scale" (e.g., LRU or model management) in the future work.
>
> [1] van Renen et al. "Why TPC Is Not Enough: An Analysis of the Amazon Redshift Fleet", VLDB'25

---

### Author Response · Authors · 2025-11-20
**Thank you, Reviewers!**

Thank you all for thoroughly reading our paper. The thoroughness is evident, as all feedback shows a deep understanding of our work.

We break our responses as follows:

1. This response discusses themes that are common across reviewers.
2. The next response contains results due to additional experiments.
3. The one after discusses additional related work.
4. Finally, we respond to each reviewer, often pointing to the aforementined responses.

## Contribution and Novelty
All reviewers mentioned Novelty.  Previously, we listed differentiators in a scattered manner (e.g., in Related Work), but now, they are upgraded to the last piece of the Introduction: "**Our contributions are ...**"
* [N.1] We are the **first** to propose an **online-learning** method that is invariant to many SQL transformations (we are adding this sentence to the manuscript). We are aware of is [1], which treats SQL (pattern) as a string (template), therefore, "{A} and {B}" will appear different than "{B} and {A}", as well as many other equivalent transformations. However, the graph, and therefore the hash and feature extraction (= node ordering), are invariant to such transformations. **Online-learning is key**, to be able to adapt on-the-fly to query patterns particular to the database and workload, without **any upfront training**. Zero history must be observed before deployment, and warm-up is quick (see Fig.5).
* [N.2] We remind reviewers that recently-proposed models (e.g., MSCN [2], DeepDB [3], FactorJoin [4], Price [5], NeuroCard[6], FLAT[7], Sun & Li [10], Woltmann et al [12]) are all trained offline and are **not** deployed in-practice, as they would need to be trained or fine-tuned to get good performance. We present the first practical learning-based solution, using a kernel-across-graphs (Eq.6). The practical implication is very fast modeling (canonical hashing & featurization of DAGs, followed by simple models).
* [N.3] While local methods are decade-old, however, methods where locality function is expressed on pairs of graphs are rare.
* [N.4] Our method (graph kernel, canonical node ordering & feature extraction), theorems, and proofs are general to any Directed Acyclic Graph (DAG) application.

## System, Scale, and Efficiency
We address issues raised by Reviewer zkyh [raised W1, W3, W4] and Reviewer AwTu [raised Weakness 2].
We now highlight in the experimental section that our **37s overhead time** running the 5k queries (46928 subqueries) on IMDb is broken as:

|Step|Time per sub-query(s)|
|-----|-- |
| hash and featurize DAG|$3*10^{-5}$|
|Model inference|$2*10^{-5}$|
|Update model (e.g., Decision-Forest)|$7*10^{-4}$|

Note: Our main contribution is an algorithm that enables online-learning across DAGs. Luckily, open-source implementations (e.g., of decision trees or linear regression) are fast and are usable given our kernel construction. *If* we want to wear an optimization hat, we could have:
* Fit the model on a different thread, saving up-to 34 seconds from the 37 seconds from the main thread.
* Use approximate nearest neighbors instead of KNN (for RBF model); or incrementally update the decision trees instead of always learning them from scratch.

However, squeezing every second of overhead is outside our interest, as we stop as soon as the algorithm is practical.
## Suboptimal writing.
* [P.1] **"LiteCard is only mentioned much later, and is intermixed with Ours."** We have improved the manuscript already to mention "LiteCard" starting from abstract & intro.
* [P.2] Fig5's caption points to a renamed ref (Reviewers **zkyh** & **FAp8**). Thanks :) we fixed it!
## Clarification on Baselines and Related Works
We have run additional experiments using a GNN (as requested by reviewer **FAp8**) on CardBench (as pointed-out by Reviewer **zkyh**). See the Additional Experiments Section below.
## bib
* [1] Malik et al, *A black-box approach to query cardinality estimation*, CIDR'07.
* [2] Kipf et al, *Learned cardinalities: Estimating correlated joins with deep learning*, CIDR'19.
* [3] Hilprecht et al, *DeepDB: Learn from Data, not from Queries!*. VLDB'20.
* [4] Wu et al, *FactorJoin: A New Cardinality Estimation Framework for Join Queries.*. SIGMOD'23.
* [5] Zeng et al, *PRICE: A Pretrained Model for Cross-Database Cardinality Estimation*, VLDB'25.
* [6] Yang et al, *NeuroCard: One Cardinality Estimator for All Tables*, VLDB'21.
* [7] Zhu et al, *FLAT: Fast, Lightweight and Accurate Method for Cardinality Estimation*, VLDB'21.
* [8] Negi et al, *Cost-guided cardinality estimation: Focus where it matters*, ICDE-W'20.
* [9] Negi et al, *Flow-Loss: Learning Cardinality Estimates That Matter*, VLDB'21.
* [10] Sun & Li, *An end-to-end learning-based cost estimator*, VLDB'20.
* [11] Sun et al, *Learned cardinality estimation: A design space exploration and a comparative evaluation*, VLDB'22.
* [12] Woltmann et al, *Cardinality estimation with local deep learning models*, aiDM-W'19.

---

> ### Author Response · Authors · 2025-11-20
> **Additional Experiments and Observations**
>
> Following Reviewer **zkyh** and Reviewer **FAp8**,  we made many GNN runs and presenting the best architecture, over many CardBench databases and workloads, trained with R.M.S.E: $||GNN(g) - y||$ and also R.M.S.log.E $|| log(1 + GNN(g)) - log(1 + y) ||$ -- the second is empirically better.
>
> # Setup
>
> In this setup, we want to
>
> * Have a standard train/test split. Each column (below) uses a different proportion for training data (25%, 50%, 75%). For our methods, we make inference on test sample **using only training samples** -- not the usual "online setup", for an **apples-to-apples** comparison.
> * We implemented a GNN architecture that is similar to GCN (with straightforward extension to heterogeneous graphs).
> * Each cell for GNN takes about 5-to-10 minutes. For our method, it takes a second.
>
> In general, we see:
>
> * log(GNN) is much better than GNN, and is competitive with LiteCARD(RBF) and LiteCARD(DF) [meaning **if we always trust ours** (always generate a prediction even when there is no history data)]
> * Ours -- LiteCARD(DF)[D >= 5] and LiteCARD(DF)[D >= 10] -- is much better , **if we selectively trust it** (when it has seen >5 or >10 data points per hash bucket).
> * These experiments further motivate Section "3.5 Hierarchical Data Structure".
>
> We will add the results below to the manuscript.
>
> # Results
>
>  ##  Dataset: stackoverflow, Qerr @ 50th %ile
> |Train Ratio (%)|25|50|75|
> |--|--|--|--|
> | GNN|2.88|2.54|2.58 |
> | log(GNN)|2.39|2.00|1.87 |
> | LiteCARD(RBF)|1.84|1.51|1.44 |
> | LiteCARD(DF)|1.80|1.46|1.33 |
> | LiteCARD(DF) [D >= 5]|1.04|1.03|1.03 |
> | LiteCARD(DF) [D >= 10]|1.04|1.02|1.02 |
>
>  ##  Dataset: airline, Qerr @ 50th %ile
> |Train Ratio (%)|25|50|75|
> |--|--|--|--|
> | GNN|2.41|2.30|2.35 |
> | log(GNN)|2.05|1.77|1.69 |
> | LiteCARD(RBF)|3.45|3.08|2.85 |
> | LiteCARD(DF)|3.45|3.07|2.82 |
> | LiteCARD(DF) [D >= 5]|1.02|1.02|1.03 |
> | LiteCARD(DF) [D >= 10]|1.00|1.00|1.01 |
>
>  ##  Dataset: accidents, Qerr @ 50th %ile
> |Train Ratio (%)|25|50|75|
> |--|--|--|--|
> | GNN|1.65|1.63|1.50 |
> | log(GNN)|1.74|1.51|1.44 |
> | LiteCARD(RBF)|1.19|1.13|1.11 |
> | LiteCARD(DF)|1.11|1.06|1.03 |
> | LiteCARD(DF) [D >= 5]|1.00|1.00|1.00 |
> | LiteCARD(DF) [D >= 10]|1.00|1.00|1.00 |
>
>  ##  Dataset: cms, Qerr @ 50th %ile
> |Train Ratio (%)|25|50|75|
> |--|--|--|--|
> | GNN|3.22|2.60|2.25 |
> | log(GNN)|2.10|1.81|1.51 |
> | LiteCARD(RBF)|2.22|1.68|1.45 |
> | LiteCARD(DF)|2.21|1.62|1.42 |
> | LiteCARD(DF) [D >= 5]|1.01|1.00|1.01 |
> | LiteCARD(DF) [D >= 10]|1.00|1.00|1.00 |
>
>  ##  Dataset: geo, Qerr @ 50th %ile
> |Train Ratio (%)|25|50|75|
> |--|--|--|--|
> | GNN|5.86|5.62|178.71 |
> | log(GNN)|1.97|1.66|1.58 |
> | LiteCARD(RBF)|1.70|1.46|1.37 |
> | LiteCARD(DF)|1.67|1.40|1.31 |
> | LiteCARD(DF) [D >= 5]|1.06|1.05|1.06 |
> | LiteCARD(DF) [D >= 10]|1.04|1.03|1.03 |
>
>  ##  Dataset: employee, Qerr @ 50th %ile
> |Train Ratio (%)|25|50|75|
> |--|--|--|--|
> | GNN|2.28|2.11|2.04 |
> | log(GNN)|1.41|1.19|1.21 |
> | LiteCARD(RBF)|1.07|1.06|1.06 |
> | LiteCARD(DF)|1.02|1.01|1.01 |
> | LiteCARD(DF) [D >= 5]|1.01|1.01|1.01 |
> | LiteCARD(DF) [D >= 10]|1.01|1.01|1.01 |
>
>  ##  Dataset: stackoverflow, Qerr @ 75th %ile
> |Train Ratio (%)|25|50|75|
> |--|--|--|--|
> | GNN|64.42|57.20|65.98 |
> | log(GNN)|5.03|3.51|3.17 |
> | LiteCARD(RBF)|6.35|5.11|4.48 |
> | LiteCARD(DF)|6.14|4.98|4.24 |
> | LiteCARD(DF) [D >= 5]|1.19|1.16|1.16 |
> | LiteCARD(DF) [D >= 10]|1.14|1.09|1.07 |
>
>
>  ##  Dataset: airline, Qerr @ 75th %ile
> |Train Ratio (%)|25|50|75|
> |--|--|--|--|
> | GNN|92.80|61.00|53.30 |
> | log(GNN)|4.44|3.18|2.87 |
> | LiteCARD(RBF)|20.14|11.90|10.69 |
> | LiteCARD(DF)|20.16|12.04|11.14 |
> | LiteCARD(DF) [D >= 5]|1.18|1.16|1.22 |
> | LiteCARD(DF) [D >= 10]|1.05|1.08|1.09 |
>
>  ##  Dataset: accidents, Qerr @ 75th %ile
> |Train Ratio (%)|25|50|75|
> |--|--|--|--|
> | GNN|3.72|3.46|3.45 |
> | log(GNN)|3.01|2.38|1.91 |
> | LiteCARD(RBF)|2.77|2.39|2.25 |
> | LiteCARD(DF)|2.53|2.25|2.05 |
> | LiteCARD(DF) [D >= 5]|1.03|1.05|1.09 |
> | LiteCARD(DF) [D >= 10]|1.02|1.01|1.01 |
>
>  ##  Dataset: cms, Qerr @ 75th %ile
> |Train Ratio (%)|25|50|75|
> |--|--|--|--|
> | GNN|166974.56|58914.54|552.22 |
> | log(GNN)|4.00|2.98|2.20 |
> | LiteCARD(RBF)|5.74|6.13|6.12 |
> | LiteCARD(DF)|5.74|6.13|6.13 |
> | LiteCARD(DF) [D >= 5]|1.14|1.09|1.09 |
> | LiteCARD(DF) [D >= 10]|1.08|1.04|1.05 |
>
>  ##  Dataset: geo, Qerr @ 75th %ile
> |Train Ratio (%)|25|50|75|
> |--|--|--|--|
> | GNN|28.12|24.89|1866031.25 |
> | log(GNN)|3.28|2.52|2.37 |
> | LiteCARD(RBF)|4.45|3.61|3.13 |
> | LiteCARD(DF)|4.29|3.51|2.98 |
> | LiteCARD(DF) [D >= 5]|1.23|1.30|1.40 |
> | LiteCARD(DF) [D >= 10]|1.14|1.09|1.11 |
>
>  ##  Dataset: employee, Qerr @ 75th %ile
> |Train Ratio (%)|25|50|75|
> |--|--|--|--|
> | GNN|655.59|16.55|12.04 |
> | log(GNN)|1.96|1.39|1.39 |
> | LiteCARD(RBF)|1.29|1.26|1.25 |
> | LiteCARD(DF)|1.16|1.11|1.11 |
> | LiteCARD(DF) [D >= 5]|1.09|1.08|1.09|
> | LiteCARD(DF) [D >= 10]|1.06|1.06|1.08|

---

> > ### Author Response · Authors · 2025-11-20
> > **Additional Related Work**
> >
> > Reviewer **FAp8** asked for comparison with baselines - Flowloss [8,9], Sun & Li [10], Sun et al [11] - and clarification on Woltmann et al [12].
> >
> >   * Flow-Loss [8,9] targets a different point in the design space than LiteCard, and is therefore not a direct baseline. It aims to improve how a single global neural cardinality model is trained by introducing an optimizer-aware loss function (instead of Q-Error) on a large offline benchmark. In contrast, LiteCard does not propose a new loss; we change the estimator integration: LiteCard leverages the highly repetitive (sub)queries patterns and trains light-weight models online from executed queries, without pre-computed labeled data. Thus, Flow-Loss is largely orthogonal to our contribution: it provides a better loss for training heavy-weight global models, while LiteCard shows that lightweight, online, local models can deliver substantial end-to-end gains at near-zero overhead. We now update this into the related work section.
> >   * Sun & Li [10] is a query-driven method that uses tree-based models (e.g. tree-LSTM and Tree-NN) to predict the cost of query plan trees. We already cited Sun & Li in the related work section.
> >   * Sun et al [11] present a design-space exploration and evaluation of learned cardinality estimators; this is a survey/benchmark paper, not a new method to compare against. Our experiments already include representative methods from the main categories they identify (query-driven MSCN [2], and data-driven DeepDB/FactorJoin [3,4]), so LiteCard is positioned as exploring a previously under-emphasized corner of that design space: online, workload-driven, local models with negligible overhead. We now add this paper into the related work section.
> >   * Woltmann et al [12] is also a local query-driven approach, but at the query table level: they train neural network models offline for the queries that share the same join tables (correspond to our H3 in the Section 3.5 Hierarchical Data Structure). However, they still need to gather large amounts of training data ahead, eg. They use 90k queries to train a local model. In contrast, LiteCard enables online learning that can run from cold-start, without upfront training data. We now clarify these distinctions in the revised related-work discussion.
> >
> >
> > Reviewer **zkyh** asked for additional comparison with data-driven methods - NeuroCard [6] and FLAT [7]. They are both data-driven generative estimators that learn a global joint distribution over the data (e.g., autoregressive models and FSPNs), require substantial offline training, and have already been extensively compared against DeepDB and related work. In our experiments, we choose DeepDB [3] and FactorJoin [4] as representative data-driven baselines: DeepDB is a well-established strong baseline, and FactorJoin is a recent method that improves end-to-end performance over earlier data-driven approaches. For example, Figure 6 in the [paper](https://arxiv.org/pdf/2212.05526) shows FactorJoin outperforming FLAT [7] and other data-driven baselines on end-to-end time, while Table 2 in FLAT [paper](https://www.vldb.org/pvldb/vol14/p1489-zhu.pdf) [7] reports FLAT outperforming NeuroCard [6] on IMDb. Thus, prior results indicate that DeepDB and FactorJoin together cover the same region of the accuracy–overhead frontier as NeuroCard and FLAT. We have updated the related-work section to cite NeuroCard and FLAT explicitly.

---

### Author Response · Authors · 2025-11-27
**Updated PDF**

We have uploaded a PDF with several changes, based on your feedback -- thank you for substantially improving our work!

* We improved the abstract -- to mention our motivation and model name LITECARD
* We added **Our main contributions** at the end of Introduction
* For motivation: We cited (in intro) van Renen et al. (2024), who show that >95% of queries repeat in same template within a month.
* We added more experiments (subsections and figures).
* We simplified the text of Experiments, and moved the baseline descriptions to appendix (to make space for the above)
* We added Related Work that was pointed-out by reviewers.

---

### Author Response · Authors · 2025-12-02
**Thank you Reviewers, Hello AC!**

First of all: Thank you reviewers who spent the time to read our paper and post detailed reviews -- you all helped progress our work! Additionally, we want to thank the reviewers (**FAp8** and **zkyh**) who followed-up on responses before the data leak bug happened -- thank you for your prompt follow-up and for upgrading your scores.

Area Chairs, we will do our best to summarize reviews and our actions using as few words as possible, to reduce your cognitive load as now you have to dig deep into several papers and be able to make an informed decision. This seems like an intense job. We hope it goes smooth for you -- and thank you, on behalf of the academic community.

## Main points and Responses

* All reviewer feedback was incorporated as paper improvements, marked in **blue**
* Reviewers asked about novelty: "What is your contribution?"
  * We have added a paragraph at end of Introduction: "**Our main contributions are**".
* Reviewers complained mentioning that our method is applicable **only when** patterns of SQL commands repeat.
  * This observation is correct. Luckily, it is the case in practice, that most queries repeat in template (as most queries are generated programmatically). We cited van Renen et al. (2024) in second-to-last paragraph of intro, who conducted a study on quantifying  repeats of query patterns.
* Reviewers asked us to compare under a "standard setup" [same train test for all methods, rather than "online learning" that allowed our method to learn from every previous example], and asked us to compare with Graph Neural Network (GNN) baselines, and on CardBench datasets
  * We now have additional standard setup (train:test) experiments, using GNN baselines, on CardBench, all in **new subsection 4.2**, with associated **Figure 11**.
* Reviewers asked us to include additional related work
  * We have added several papers to **Related Work section**.
* Some reviewers thought that some aspects of the work are not straightforward to understand.
  * We added some example in textual responses over here (you may search "the steps are" within this page). We plan to make a figure.
* Minor typos:
  * All improved.

---

### Meta-Review · Area_Chair_28Xa · 2025-12-25

**Summary:**

This paper presents a novel online cardinality estimation framework, referred to as LITECARD which is designed to balance estimation accuracy with practical system efficiency. The main idea is to decompose the cardinality estimation problem into many small, localized regression models, each specialized for a specific subquery pattern identified via graph isomorphism hashing. The paper seems to be a good systems paper that uses ML as a tool. However, in the context of ML, the contribution is not so clear yet. There are critical concerns in the limited novelty, unclear description of the pipeline, and scalability issues. I believe the paper fits DB conferences (VLDB, SIGMOD) better,
since the primary contribution, methodology, and evaluation metrics are deeply rooted in database systems research. Therefore, the paper is not recommended for acceptance in its current form. I hope authors found the review comments informative and can improve their paper by addressing these carefully in future submissions.

**Reviewer Concerns:**

The authors have done a nice job addressing most of the reviewers' concerns. However, from partial information available for me, some reviewers were not fully satisfied.

**Reviewer Scores:**

I expect most reviewers will maintain their original score.

---

### Decision · Program_Chairs · 2026-01-26

Reject